# Medix: Out-of-Distribution Detection from Unlabeled Wild Data via Robust Gradient Statistics

**Momin Abbas** *momin.abbas1@ibm.com*
*IBM*

**Ali Falahati** *afalahat@uwaterloo.ca*
*University of Waterloo*

**Hossein Goli** *hossein.goli@cuhk.edu.hk*
*CUHK*

**Mohammad Mohammadi Amiri** *mamiri@rpi.edu*
*Rensselaer Polytechnic Institute*

**Reviewed on OpenReview:** *https://openreview.net/forum?id=jFjA24PBJx*

## Abstract

Out-of-distribution (OOD) detection plays a crucial role in ensuring the robustness of machine learning systems deployed in real-world applications. Recent approaches have explored the use of unlabeled data, showing potential for enhancing OOD detection capabilities. However, effectively utilizing unlabeled in-the-wild data remains challenging due to the mixed nature of both in-distribution (InD) and OOD samples. The lack of a distinct set of OOD samples complicates the task of training an optimal OOD classifier. In this work, we introduce Medix, a novel framework designed to identify potential outliers from unlabeled data using the median-based robust gradient statistics. We use the median because it provides a stable estimate of the central tendency, as an OOD detection mechanism, due to its robustness against noise and outliers. Using these identified outliers, along with labeled InD data, we train a robust OOD classifier. From a theoretical perspective, we derive error bounds that demonstrate Medix achieves a low error rate. Empirical results further substantiate our claims, as Medix outperforms existing methods across the board in open-world settings.

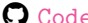 Code

## 1 Introduction

Deploying machine learning models in real-world applications often exposes them to challenges related to safety and reliability, particularly due to the presence of out-of-distribution (OOD) data. These OOD samples, which come from unknown categories, should not be predicted by the model. However, neural networks are inherently vulnerable and lack the necessary mechanisms to detect and handle OOD inputs in practice (Nguyen et al., 2015).

Identifying OOD samples during inference is critical yet not easy, as models are not exposed to unknown distributions during training and, therefore, cannot reliably distinguish OOD from in-distribution (InD) data. To address this challenge, recent approaches (Katz-Samuels et al., 2022; Du et al., 2024a) have explored using additional "in-the-wild" data to improve OOD detection. Specifically, Katz-Samuels et al. (2022) introduced a method that uses unlabeled wild data for regularizing model training, while still focusing on classifying labeled InD data. The advantage of using such unlabeled wild data lies in its availability, being easily collectible once a model is deployed in its operating environment. This approach allows the model to better capture the true distribution of OOD data encountered during test time and leads to a more robust OOD detection.

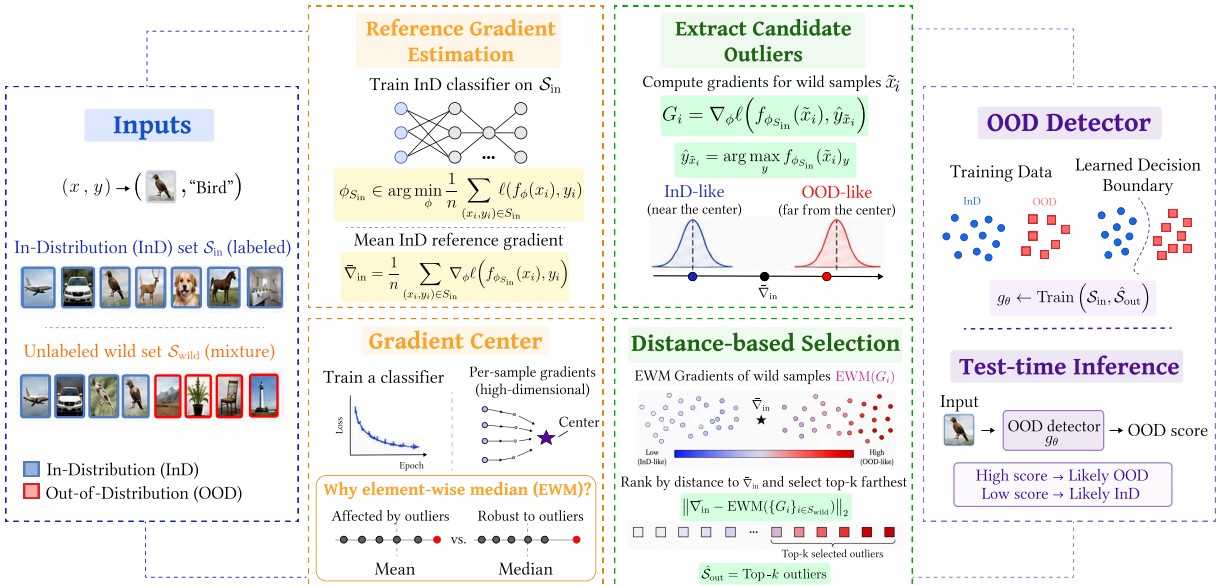

Figure 1: Overview of Medix. The method first trains an InD classifier to compute a reference gradient from labeled InD data. It then extracts candidate OOD samples from unlabeled wild data by iteratively identifying gradients that maximally reduce the deviation between the element-wise median (EWM) and the InD reference gradient. Finally, the extracted candidate outliers together with the labeled InD data are used to train an OOD detector.

However, leveraging unlabeled wild data presents challenges due to the complex mixture of InD and OOD data. The absence of a distinct and clean set of OOD samples complicates the development of robust OOD detection methods, especially since the OOD detector model only encounters data drawn from this mixed distribution, without knowledge of whether each sample is from the InD or OOD category. Currently, the problem remains underexplored, with substantial opportunities for further progress. Moreover, few studies establish a formal theoretical foundation, and to the best of our knowledge, Du et al. (2024a) is the only work that provides such a foundation for the "in-the-wild" setting.

Meanwhile, recent studies have demonstrated the effectiveness of median-based approaches in data pruning (Acharya et al., 2024). Motivated by these developments, this paper aims to answer the following question:

> 💡 **Question.** Can robust gradient statistics leverage unlabeled wild data to facilitate OOD detection, despite the data containing an unknown mixture of InD and OOD samples?

To answer this question, we propose Medix (Figure 1), a framework that converts unlabeled wild data into useful supervision for OOD detection by extracting candidate OOD samples from the unlabeled mixture. We show that this filtering procedure admits provable error guarantees and leads to robust OOD detectors in practice. We provide theoretical guarantees for minimal error, which is further validated through experiments. We benchmark our approach against two categories of methods: (1) those trained solely on InD data, and (2) those trained with both InD data and an auxiliary unlabeled dataset. On CIFAR-100 (Krizhevsky et al., 2009), Medix demonstrates a significant improvement over the strong baseline KNN+ (Sun et al., 2022), outperforming it by an average of 40.98% in terms of FPR95. Unlike approaches such as Outlier Exposure (Hendrycks et al., 2019), which rely on a clean, auxiliary unlabeled dataset (i.e. they make a strong distributional assumption that the auxiliary data is completely separable from the InD data), Medix achieves superior results without such assumptions, offering greater flexibility. Compared to WOODS (Katz-Samuels et al., 2022), Medix reduces the average FPR95 by 1.32% on CIFAR-100 and 2.60% on CIFAR-10.

Our contributions are as follows:

**C1)** We propose Medix, a median-centric approach that filters outliers from the unlabeled wild data and then trains an OOD detector on the identified outliers and the InD samples. Our main contribution is the filtering stage.

**C2)** We establish theoretical guarantees for the robustness of median-based filtering in identifying both inliers and outliers within unlabeled mixtures.

**C3)** We conduct an extensive evaluation of Medix across eleven InD-OOD pairs, comparing its performance against 20 competitive baselines. Our results demonstrate that Medix outperforms all the baselines, achieving superior performance across the board.

## 2 Preliminaries and Problem Setup

In this section, we provide an overview of the OOD detection problem, and then formally define the data setup, model architecture, loss functions, and the learning goal.

**Labeled In-Distribution Data.** Consider the input space $\mathcal{X}$ and the label space $\mathcal{Y} = \{1, \ldots, K\}$, which together define the structure of InD data. A labeled dataset $\mathcal{S}_{\text{in}} = \{(\boldsymbol{x}_1, y_1), \ldots, (\boldsymbol{x}_n, y_n)\}$ is generated by sampling $n$ pairs independently and identically distributed (i.i.d.) from $\mathbb{P}_{XY}$, an unknown joint probability distribution over $\mathcal{X} \times \mathcal{Y}$. The marginal distribution of $\mathbb{P}_{XY}$ on $\mathcal{X}$ is denoted as $\mathbb{P}_{\text{in}}$, representing the underlying distribution of InD inputs. We use $\mathcal{S}_{\text{in}}$ to train an InD model.

**Out-of-distribution detection.** We address a practical scenario where the model is trained using labeled InD data but is later deployed in environments that may contain OOD inputs from classes not represented in the training data, i.e., for some label $y \notin \mathcal{Y}$. The model is expected to abstain from making predictions for such OOD inputs. At inference time, the primary objective is to determine whether a given input belongs to the InD distribution or arises from an OOD source.

**Unlabeled wild data.** One of the primary obstacles in OOD detection is the scarcity of labeled OOD samples. The potential sample space of OOD data can be very large, making the collection of labeled examples both costly and impractical. To address this, we introduce unlabeled wild data, $\mathcal{S}_{\text{wild}} = \{\tilde{\boldsymbol{x}}_1, \ldots, \tilde{\boldsymbol{x}}_m\}$, into our learning framework to better mimic real-world scenarios as proposed by Katz-Samuels et al. (2022). Wild data is a blend of InD and OOD samples and can be readily collected during the deployment phase of a pre-trained model on $\mathcal{S}_{\text{in}}$. Similar to Du et al. (2024a); Katz-Samuels et al. (2022), we adopt the Huber contamination model (Huber, 1964) to characterize the marginal distribution of the wild data

$$\mathbb{P}_{\text{wild}} := (1 - \pi)\mathbb{P}_{\text{in}} + \pi\mathbb{P}_{\text{out}}, \quad \pi \in (0, 1], \tag{1}$$

where $\pi$ denotes the proportion of contamination and $\mathbb{P}_{\text{out}}$ captures the OOD distribution on $\mathcal{X}$. We note that the scenario where $\pi = 0$ corresponds to the absence of OOD samples, rendering the problem trivial.

**Models and Loss Functions.** Let $f_\phi : \mathcal{X} \to \mathbb{R}^K$ represent the InD classifier parameterized by $\phi \in \Phi$, where $\Phi$ denotes the parameter space for this classifier. The output of $f_\phi$ corresponds to a soft probability distribution over the $K = |\mathcal{Y}|$ InD classes. The loss function for the labeled InD data is defined as $\ell : \mathbb{R}^K \times \mathcal{Y} \to \mathbb{R}$. For OOD detection, we introduce a separate classifier $g_\theta : \mathcal{X} \to \mathbb{R}$, parameterized by $\theta \in \Theta$, with $\Theta$ as the parameter space. The binary loss function associated with $g_\theta$ is denoted as $\ell_b(g_\theta(x), y_b)$, where $y_b \in \mathcal{Y}_b := \{y_+, y_-\}$. Here, $y_+ > 0$ represents the InD class, while $y_- < 0$ corresponds to the OOD class.

**Learning objective.** Our learning framework is designed to simultaneously train the OOD detector $g_\theta$ and the multi-class classifier $f_\phi$, leveraging both the InD data $\mathbb{P}_{\text{in}}$ and the wild data $\mathbb{P}_{\text{wild}}$. During testing, we evaluate the performance using the following metrics:

$$\downarrow \text{FPR}(g_\theta) = \mathbb{E}_{\boldsymbol{x} \sim \mathbb{P}_{\text{out}}^{\text{test}}} \left[ \mathcal{I}\{g_\theta(\boldsymbol{x}) = \text{in}\} \right],$$
$$\uparrow \text{TPR}(g_\theta) = \mathbb{E}_{\boldsymbol{x} \sim \mathbb{P}_{\text{in}}} \left[ \mathcal{I}\{g_\theta(\boldsymbol{x}) = \text{in}\} \right],$$
$$\uparrow \text{Acc}(f_\phi) = \mathbb{E}_{(\boldsymbol{x},y) \sim \mathbb{P}_{XY}} \left[ \mathcal{I}\{f_\phi(\boldsymbol{x}) = y\} \right],$$

where $\mathcal{I}\{\cdot\}$ denotes the indicator function, and $\mathbb{P}_{\text{out}}^{\text{test}}$ represents the OOD test data distribution.

## 3 Medix: Median-Centric Framework for OOD Detection

In this section, we present a novel learning paradigm, termed **Medix**, designed for OOD detection by harnessing the power of unlabeled wild data. Our framework overcomes the limitations of conventional approaches that rely exclusively on InD data and is particularly well-suited for applications in open-world environments, where models are often confronted with previously unseen inputs. The Medix framework is composed of two stages: **1) Outlier Extraction**: A filtering process that isolates candidate OOD samples from the unlabeled wild data (explained in Section 3.1), and **2) Detector Training**: train a binary OOD detector using both InD data and the outlier candidates identified in the previous step (explained in Section 3.2). For stage 2, we follow the protocol introduced by Du et al. (2024a). As we will demonstrate in the subsequent sections, this two-step methodology not only facilitates the effective extraction of OOD data from the unlabeled wild data, but also establishes a robust foundation for deploying machine learning models in dynamic, open-world scenarios.

### 3.1 Extracting Candidate Outliers from the Wild Data

To isolate potential outliers from the wild mixture $\mathcal{S}_{\text{wild}}$, our framework leverages an optimization-based approach that exploits the gradients of the model parameters. These gradients are derived from a classification model, $f_\phi$, which is trained solely on the InD dataset $\mathcal{S}_{\text{in}}$.

**Reference gradient estimation from InD data.** The first step in our proposed framework is to estimate a reference gradient using the InD dataset $\mathcal{S}_{\text{in}}$. This is achieved by training a classifier $f_\phi$ on $\mathcal{S}_{\text{in}}$ through empirical risk minimization (ERM) as follows

$$\phi_{\mathcal{S}_{\text{in}}} \in \arg\min_{\phi \in \Phi} \mathcal{L}_{\mathcal{S}_{\text{in}}}(f_\phi), \quad \text{where} \quad \mathcal{L}_{\mathcal{S}_{\text{in}}}(f_\phi) = \frac{1}{n} \sum_{(\boldsymbol{x}_i, y_i) \in \mathcal{S}_{\text{in}}} \ell(f_\phi(\boldsymbol{x}_i), y_i), \tag{2}$$

where $\phi_{S_{\text{in}}}$ denotes the learned parameters. Once the classifier has been trained, we compute the mean gradient $\bar{\nabla}_{\text{in}}$ as the average of the gradients of the loss function with respect to the model parameters over the InD data:

$$\bar{\nabla}_{\text{in}} = \frac{1}{n} \sum_{(\boldsymbol{x}_i, y_i) \in \mathcal{S}_{\text{in}}} \nabla \ell(f_{\phi_{\mathcal{S}_{\text{in}}}}(\boldsymbol{x}_i), y_i). \tag{3}$$

In our approach, $\bar{\nabla}_{\text{in}}$ serves as the reference gradient, allowing the quantification of deviations for other data points relative to this reference.

**Motivation.** We hypothesize that increasing the number of OOD samples in the wild dataset, $\mathcal{S}_{\text{wild}}$, will lead to a greater deviation from the average InD gradient, $\bar{\nabla}_{\text{in}}$. To test this hypothesis, we design an initial experiment using CIFAR-10 (Krizhevsky et al., 2009) as the InD dataset and SVHN (Netzer et al., 2011) as the OOD dataset. Specifically, $\mathcal{S}_{\text{wild}}$ consists of 10,000 samples drawn from CIFAR-10, ensuring that these samples are disjoint from the training set used to train the model $\phi_{\mathcal{S}_{\text{in}}}$, which we leverage to compute $\bar{\nabla}_{\text{in}}$. We incrementally add SVHN OOD samples to $\mathcal{S}_{\text{wild}}$ and track the behavior of the $L_2$-norm deviation between $\bar{\nabla}_{\text{in}}$ and the element-wise median (EWM) of the gradients of the wild dataset as follows:

$$\left\| \bar{\nabla}_{\text{in}} - \text{EWM}\left( \left\{ \nabla \ell \left( f_{\phi_{\mathcal{S}_{\text{in}}}}(\tilde{\boldsymbol{x}}_i), \hat{y}_{\tilde{\boldsymbol{x}}_i} \right) \right\}_{i \in \mathcal{S}_{\text{wild}}} \right) \right\|. \tag{4}$$

Here EWM(·) denotes the element-wise median function, and $\hat{y}_{\tilde{\boldsymbol{x}}_i}$ represents the predicted label for a wild sample $\tilde{\boldsymbol{x}}_i$. Our results, depicted in Figure 2, reveal a clear and monotonic increase in the $L_2$-norm deviation, supporting our hypothesis. This observation serves as a key motivation for the method we introduce in the subsequent section. Notably, the stopping criterion for our algorithm is derived from this monotonically increasing behavior, where we terminate the algorithm when the $L_2$-norm deviation between consecutive iterations drops below a threshold $\epsilon$; we will explain this method in detail in the following section.

**Filtering potential outliers from unlabeled wild data.** Motivated by the results in Figure 2, we formulate the following optimization problem to identify the outlier subset $\mathcal{S}_{\text{out}}^*$ in $\mathcal{S}_{\text{wild}}$:

$$\mathcal{S}_{\text{in}}^* = \arg\min_{\mathcal{S} \subseteq \mathcal{S}_{\text{wild}}} \left\| \bar{\nabla}_{\text{in}} - \text{EWM}(G_{\mathcal{S}}) \right\|, \quad \text{where} \quad G_{\mathcal{S}} = \left\{ \nabla \ell \left( f_{\phi_{\mathcal{S}_{\text{in}}}}(\tilde{\boldsymbol{x}}_i), \hat{y}_{\tilde{\boldsymbol{x}}_i} \right) \right\}_{i \in \mathcal{S}}. \tag{5}$$

The mean gradient is the element-wise average over InD samples, and the EWM is the per-coordinate median over wild samples, computed along the sample axis (for details see Appendix A.11).

The above optimization problem aims to find a subset $\mathcal{S}$ in $\mathcal{S}_{\text{wild}}$ that minimizes the distance between the EWM of the gradients and the average gradient $\bar{\nabla}_{\text{in}}$. According to Figure 2, such a subset, denoted by $\mathcal{S}_{\text{in}}^*$, may well represent the InD data in $\mathcal{S}_{\text{wild}}$, in which case $\mathcal{S}_{\text{out}}^* = \mathcal{S}_{\text{wild}} \backslash \mathcal{S}_{\text{in}}^*$ capture the OOD in $\mathcal{S}_{\text{wild}}$.

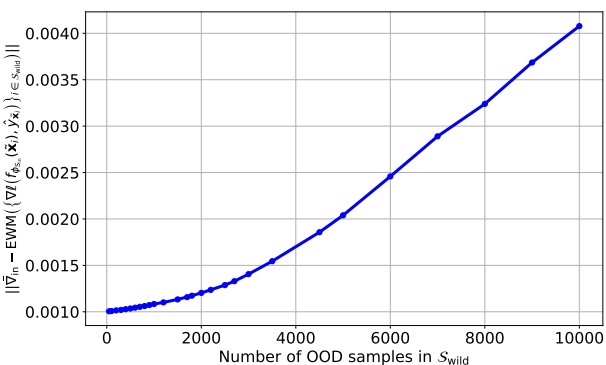

Figure 2: Distance deviation as we increase OOD samples in $\mathcal{S}_{\text{wild}}$.

Solving the optimization problem in 5 is computationally intractable. To address this, we propose a greedy approximation based on a leave-one-out approach, as outlined in Algorithm 1. The algorithm implements an iterative procedure for outlier detection from a wild dataset, leveraging deviations in gradient information relative to the InD dataset.

The algorithm begins by computing the EWM of the wild data gradients at each iteration, which serves as a reference for comparing against the average gradient of the InD data $\bar{\nabla}_{\text{in}}$. We denote by $d_t$, the $L_2$ distance between the average InD gradient $\bar{\nabla}_{\text{in}}$ and the EWM gradients of the data left in the wild set, represented by $\mathcal{S}$, i.e., $d_t = \|\text{EWM}(G_\mathcal{S}) - \bar{\nabla}_{\text{in}}\|$. The algorithm then iteratively identifies samples in $\mathcal{S}$ that incur the most significant drop in the $L_2$ distance with $d_t$ when removed from $\mathcal{S}$ as OOD. Specifically, having data samples $\mathcal{S}$ left, we remove each sample $i \in \mathcal{S}$ and compute the EWM as $\text{EWM}(G_{\mathcal{S} \backslash \{i\}})$ and the distance to $\bar{\nabla}_{\text{in}}$ as $\|\text{EWM}(G_{\mathcal{S} \backslash \{i\}}) - \bar{\nabla}_{\text{in}}\|$. We then find the drop in distance $\delta_i = d_t - \|\text{EWM}(G_{\mathcal{S} \backslash \{i\}}) - \bar{\nabla}_{\text{in}}\|$ and find the $k$ samples with the largest drop and identify them as OOD. The algorithm repeats until there is no significant drop in $\delta_i$ or a maximum number of iterations is reached. The convergence criterion is based on the change in the $L_2$ distance between two iterations, which must fall below a predefined $\epsilon$ threshold; this criterion is inspired by the monotonically increasing trend that we observed in our preliminary experiment in Figure 2, i.e., as a large number of OOD samples are removed, the $L_2$ distance gradually decreases to a small value, signaling the point at which the algorithm should halt to avoid identifying InD samples as outliers, thus preventing any degradation in performance.

---

**Algorithm 1** Iterative Outlier Detection via **Medix**

**Require:** $\bar{\nabla}_{\text{in}}$, $G_i = \nabla\ell(f_{\phi_{\mathcal{S}_{\text{in}}}}(\tilde{\boldsymbol{x}}_i), \hat{y}_{\tilde{\boldsymbol{x}}_i})$, maximum iterations $T$, hyperparameters $\epsilon$, $k$

1: Initialize $\mathcal{S} \leftarrow \mathcal{S}_{\text{wild}}$ (wild set), $\mathcal{O} \leftarrow \emptyset$ (outliers), $t \leftarrow 0$ (iteration), $d_t \leftarrow 0$, $\delta_{\max} \leftarrow \infty$ (deviations), $\mathcal{I}_k \leftarrow \emptyset$
2: **while** $t \leq T$ and $|\delta_{\max}| > \epsilon$ **do**
3:     $\mathcal{O} \leftarrow \mathcal{O} \cup \{\tilde{\boldsymbol{x}}_i : i \in \mathcal{I}_k\}$                                      ▷ Add outliers to set $\mathcal{O}$
4:     $d_t \leftarrow \|\text{EWM}(G_\mathcal{S}) - \bar{\nabla}_{\text{in}}\|$                                ▷ Compute L2 deviation
5:     **for** each $i \in \mathcal{S}$ **do**
6:         $\delta_i \leftarrow d_t - \|\text{EWM}(G_{\mathcal{S} \backslash \{i\}}) - \bar{\nabla}_{\text{in}}\|$
7:     **end for**
8:     $\mathcal{I}_k \leftarrow \text{indices}(\text{top-}k(\{\delta_i\}_{i \in \mathcal{S}}))$                          ▷ Select top-$k$ indices
9:     $\mathcal{S} \leftarrow \mathcal{S} \backslash \mathcal{I}_k$                                      ▷ Remove outliers from $\mathcal{S}$
10:    $\delta_{\max} \leftarrow \max_{i \in \mathcal{S}}\{\delta_i\}$, $t \leftarrow t + 1$
11: **end while**
12: **return** $\hat{\mathcal{S}}_{\text{out}} = \mathcal{O}$                                          ▷ Return detected outliers

---

## 3.2 Training the OOD Detector with Candidate Outliers

After identifying the candidate outlier set $\hat{\mathcal{S}}_{\text{out}}$ from the wild data, we proceed to train the OOD detector $g_\theta$ designed to maximize the distinction between InD and candidate outlier data following the protocol in Du et al. (2024a). The objective function enforces separability at the decision boundary (thresholded at

0), assigning positive outputs for labeled InD samples $\boldsymbol{x} \in \mathcal{S}_{\text{in}}$ and negative outputs for candidate outliers $\tilde{\boldsymbol{x}} \in \hat{\mathcal{S}}_{\text{out}}$. Specifically, the loss function is defined as:

$$
\begin{aligned}
\mathcal{L}_{\mathcal{S}_{\text{in}}, \hat{\mathcal{S}}_{\text{out}}}(g_\theta) &= \mathcal{L}^+_{\mathcal{S}_{\text{in}}}(g_\theta) + \mathcal{L}^-_{\hat{\mathcal{S}}_{\text{out}}}(g_\theta), \\
\mathcal{L}^+_{\mathcal{S}_{\text{in}}}(g_\theta) &= \mathbb{E}_{\boldsymbol{x} \in \mathcal{S}_{\text{in}}} \mathcal{I}\{g_\theta(\boldsymbol{x}) \leq 0\}, \\
\mathcal{L}^-_{\hat{\mathcal{S}}_{\text{out}}}(g_\theta) &= \mathbb{E}_{\tilde{\boldsymbol{x}} \in \hat{\mathcal{S}}_{\text{out}}} \mathcal{I}\{g_\theta(\tilde{\boldsymbol{x}}) > 0\}.
\end{aligned}
\tag{6}
$$

## 4 Theoretical Analysis

We now present the theoretical guarantees of Medix's filtering stage. The following theorems provide provable upper bounds on the misclassification rates for both InD and OOD points. Together, they demonstrate the two-sided robustness of our EWM filtering. For detailed proofs, see Appendix C.

**Setup.** Under the Huber contamination model in Eq. 1, each wild point $\tilde{x}_i \in \mathcal{S}_{\text{wild}}$ is independently InD with probability $1 - \pi$ and OOD with probability $\pi$. This partitions $\mathcal{S}_{\text{wild}}$ into a true InD subset $\mathcal{I}$ and a true OOD subset $\mathcal{O}$, whose (random) sizes we write $M_{\text{in}} = |\mathcal{I}|$ and $M_{\text{out}} = |\mathcal{O}|$, with $M_{\text{in}} + M_{\text{out}} = m$ and $\mathbb{E}[M_{\text{in}}] = (1 - \pi)m$. We denote by $G_i = \nabla \ell(f_{\phi_{\mathcal{S}_{\text{in}}}}(\tilde{x}_i), \hat{y}_{\tilde{x}_i}) \in \mathbb{R}^d$ the gradient of wild sample $i$, with $j$-th coordinate $G_{i,j}$ and $d$ the gradient dimension. Our analysis targets the *ideal size-constrained* EWM filter, i.e. the exact minimizer of the objective in Eq. 5 restricted to subsets of size $M_{\text{in}}$,

$$
\mathcal{S}^\star \in \arg \min_{\substack{\mathcal{S} \subseteq \mathcal{S}_{\text{wild}} \\ |\mathcal{S}| = M_{\text{in}}}} \left\| \text{EWM}(G_{\mathcal{S}}) - \bar{\nabla}_{\text{in}} \right\|_2,
\tag{7}
$$

which Algorithm 1 approximates greedily. We quantify its quality by the inlier misclassification rate, the fraction of true InD points it discards: $\text{ERR}_{\text{in}} = |\mathcal{I} \setminus \mathcal{S}^\star| / M_{\text{in}}$.

**Assumption 4.1** (Sub-Gaussian InD gradients). *The InD gradients $\{G_i\}_{i \in \mathcal{I}}$ are i.i.d., and each centered coordinate $G_{i,j} - \bar{\nabla}_{\text{in},j}$ is sub-Gaussian with variance proxy $\sigma^2$.*

---

**Theorem 4.2** (Inlier Misclassification Bound and EWM Stability). *Let Assumption 4.1 hold, fix a confidence level $\delta \in (0, 1)$, and set $\epsilon = \sigma \sqrt{2 \log(4dm/\delta)}$. If*

$$
m \geq \frac{2 \log(2/\delta)}{(1 - \pi)^2},
$$

*then, with probability at least $1 - \delta$, the ideal filter $\mathcal{S}^\star$ of Eq. 7 satisfies*

$$
\text{ERR}_{\text{in}} \leq \underbrace{\frac{2}{(1 - \pi)^2} \sqrt{\frac{\log(2/\delta)}{2m}}}_{\text{mixture-count concentration}} + \underbrace{\frac{\pi}{1 - \pi}}_{\text{unavoidable contamination budget}}.
$$

*Moreover, the selected subset satisfies the EWM stability guarantee*

$$
\left\| \text{EWM}(G_{\mathcal{S}^\star}) - \bar{\nabla}_{\text{in}} \right\|_2 \leq \epsilon \sqrt{d}.
$$

---

Theorem 4.2 gives two guarantees for the ideal size-constrained EWM filtering rule. First, the selected subset has controlled EWM deviation from the InD reference gradient: since the true InD subset $\mathcal{I}$ is feasible, optimality ensures that $\mathcal{S}^\star$ cannot have larger median deviation than $\mathcal{I}$. Second, because the filter retains $M_{\text{in}}$ samples, each retained OOD point displaces one true InD point. Thus the inlier error is controlled by the realized OOD-to-InD ratio. The concentration term comes from fluctuations in the Huber contamination count, while the contamination term captures the nominal ratio $\pi/(1 - \pi)$.

We now bound the complementary OOD-side error, namely the fraction of true OOD points that the ideal filter $\mathcal{S}^\star$ retains as InD. We write this quantity as $\text{ERR}_{\text{out}} = |\mathcal{S}^\star \cap \mathcal{O}| / M_{\text{out}}$.

**Assumption 4.3** (Sub-Gaussian OOD gradients with separation). The OOD gradients $\{G_i\}_{i \in \mathcal{O}}$ are i.i.d., and each coordinate is sub-Gaussian with variance proxy $\sigma_{\text{out}}^2$, centered at the corresponding coordinate of $\mu_{\text{out}}$. Moreover, the OOD mean is separated from the reference gradient, $\left\| \mu_{\text{out}} - \bar{\nabla}_{\text{in}} \right\|_2 \geq \Delta\sqrt{d}$ for some $\Delta > 0$.

---

**Theorem 4.4** (Outlier Misclassification Bound). *Let Assumption 4.3 hold, and suppose $0 < \pi \leq 1/2$. Fix a confidence level $\delta \in (0,1)$ and a tolerance $\epsilon \in (0, \Delta)$. Assume that the EWM of the true InD gradients in the wild set is $\epsilon$-accurate with respect to the InD reference gradient, namely*

$$\left| \text{EWM}(G_{\mathcal{I}}) - \bar{\nabla}_{\text{in}} \right|_2 < \epsilon\sqrt{d},$$

*where $\mathcal{I} \subseteq \mathcal{S}_{\text{wild}}$ denotes the true InD subset. If*

$$m \geq \frac{2\log(2/\delta)}{\pi^2},$$

*then, with probability at least $1 - \delta$, the ideal filter $\mathcal{S}^\star$ of Eq. 7 satisfies*

$$\text{ERR}_{\text{out}} \leq \underbrace{2d\exp\left(-\frac{(\Delta - \epsilon)^2}{2\sigma_{\text{out}}^2}\right)}_{separation} + \underbrace{\frac{2}{\pi^2}\sqrt{\frac{\log(2/\delta)}{2m}}}_{mixture\text{-}count\ concentration} + \underbrace{\frac{1 - \pi}{2\pi}}_{contamination}.$$

---

Theorem 4.4 focuses on the standard contamination regime $0 < \pi \leq 1/2$, where the wild sample is primarily InD but contains an unknown OOD fraction. This is the natural regime for median-based robust filtering: when OOD points form a majority, the coordinate-wise median is no longer anchored by the InD component. The theorem assumes that the EWM of the true InD gradients in the wild set is close to the InD reference gradient. The following corollary provides a sufficient high-probability condition for this reference accuracy assumption under bounded fourth moments of the true InD gradients.

---

**Corollary 4.5** (Reference accuracy under bounded fourth moments). *Suppose the true InD gradients in the wild set have bounded fourth moments around the reference gradient $\bar{\nabla}_{\text{in}}$. Then the EWM of these InD gradients concentrates around $\bar{\nabla}_{\text{in}}$ for any tolerance $\epsilon > 0$ satisfying $p_\epsilon := \mu_4/\epsilon^4 < 1/2$,*

$$\left\| \text{EWM}(G_I) - \bar{\nabla}_{\text{in}} \right\|_2 \leq \epsilon\sqrt{d}$$

*with probability at least*

$$1 - 2d\exp\left(-2M_{\text{in}}\left(\frac{1}{2} - p_\epsilon\right)^2\right).$$

---

Theorem 4.4 complements the previous result by bounding the fraction of OOD points mistakenly retained as InD. Together, Theorems 4.2 and 4.4 provide a two-sided guarantee for the robustness of the Medix filtering method. The inlier and outlier misclassification rates are governed by a balance between the following three effects:

**Contamination Effect.** For Theorem 4.2, the term $\pi/(1 - \pi)$ captures the unavoidable OOD-to-InD displacement budget. Because the ideal filter retains $M_{\text{in}}$ samples, each retained OOD point displaces one true InD point. This term is nontrivial in the standard contamination regime $\pi < 1/2$. Conversely, Theorem 4.4 includes the penalty $(1 - \pi)/(2\pi)$, reflecting the difficulty of isolating OOD points when they are underrepresented.

**Concentration Effect.** Both bounds exploit the sub-Gaussian nature of the gradient coordinates. For InD data, this ensures that most gradients concentrate near $\bar{\nabla}_{\text{in}}$, keeping the median stable even in finite samples. For OOD points, concentration around $\mu_{\text{out}}$ allows us to quantify the risk that a misaligned OOD sample

slips past the filter. These concentration terms decay as $1/\sqrt{m}$, with constants depending on the mixture proportions: $(1-\pi)^{-2}$ for the inlier bound and $\pi^{-2}$ for the outlier bound.

**Separation Effect.** Unique to Theorem 4.4, this effect quantifies how far the OOD mean gradient must lie from the InD mean gradient in order to reliably reject OOD samples. The exponential term $\exp(-(\Delta - \epsilon)^2/2\sigma_{\text{out}}^2)$ captures this trade-off: the more separated the distributions, the less likely it is for OOD gradients to fool the filter.

> **Remark 4.6.** Eq. 5 defines the ideal filtering problem, and Theorems 4.2 and 4.4 analyze its exact solution $\mathcal{S}^\star$. Algorithm 1 is our tractable approximation to this objective. Accordingly, the theorems establish the robustness of the target criterion, while Algorithm 1 is supported by empirical results.

> Theorems 4.2 and 4.4 jointly establish that, with high probability, median filtering achieves robust separation of InD and OOD samples in unlabeled data mixtures. The fraction of InD samples misclassified as outliers is bounded by contamination and concentration effects, while the fraction of OOD points incorrectly retained is governed by an exponential separation term, a concentration bound, and a reverse contamination effect.

## 5 Experiments

This section will demonstrate the efficacy of Medix across various InD-OOD dataset pairs, benchmarking it against 20 widely-used baselines. All experiments are performed on hardware equipped with NVIDIA A100-SXM4-80GB GPUs. We provide the necessary code to reproduce our results.

### 5.1 Models, Datasets, and Baselines

**Datasets.** We use the same experimental protocol as Katz-Samuels et al. (2022), which introduced the problem of learning OOD detectors with wild data. Specifically, we use CIFAR-10 and CIFAR-100 as InD datasets ($\mathbb{P}_{\text{in}}$). For OOD testing, we select a suite of image datasets, including PLACES365 (Zhou et al., 2017), SVHN, TEXTURES (Cimpoi et al., 2014), and LSUN-RESIZE & LSUN-C (Yu et al., 2015) as OOD datasets ($\mathbb{P}_{\text{out}}$). To simulate wild data ($\mathbb{P}_{\text{wild}}$), we combine a subset of InD data ($\mathbb{P}_{\text{in}}$) with the OOD data ($\mathbb{P}_{\text{out}}$) under a default mixing parameter $\pi = 0.5$. For example, when using PLACES365 as an OOD test set, we construct a wild mixture by combining CIFAR with PLACES365 as wild data and test on PLACES365 as the OOD set. This procedure is repeated across all OOD datasets and baselines. The InD CIFAR dataset is split into two halves: the first 25,000 images to train $\phi_{\mathcal{S}_{\text{in}}}$, while the remaining images to generate the wild mixture $\mathcal{S}_{\text{wild}}$. For gradient computation, we use the penultimate layer weights, as these have been shown to be particularly informative for OOD detection (Huang et al., 2021).

**Evaluation metrics.** We evaluate using three standard metrics: (1) False Positive Rate (FPR95↓) of OOD samples when the True Positive Rate of InD samples is 95%, (2) Area Under the Receiver Operating Characteristic Curve (AUROC↑), and (3) InD Classification Accuracy (InD Acc↑).

**Baselines.** Our comparison is categorized based on whether they are trained using only InD data or both InD and wild data. For the methods trained exclusively on InD data ($\mathbb{P}_{\text{in}}$), we compare Medix against a variety of established OOD detection methods, including Maximum Softmax Probability (MSP) (Hendrycks & Gimpel, 2017), ODIN (Liang et al., 2018), Mahalanobis Distance (Lee et al., 2018b), Energy Score (Liu et al., 2020c), ReAct (Sun et al., 2021), DICE (Sun & Li, 2022), KNN Distance (Sun et al., 2022), and ASH (Djurisic et al., 2023); these methods use a model trained with softmax cross-entropy loss. Additionally, we also compare against methods based on contrastive loss, such as CSI (Tack et al., 2020b) and KNN+ (Sun et al., 2022), for a more comprehensive comparison. For methods that leverage both InD and wild data, we compare against Outlier Exposure (OE) (Hendrycks et al., 2019) and energy-regularization learning (Liu et al., 2020c), which regularize the training by promoting lower confidence or higher energy on outlier data. We also include a comparison with WOODS (Katz-Samuels et al., 2022), which introduced the concept of wild unlabeled data and utilizes it for OOD detection through a constrained optimization approach. Finally,

we included more recent baselines, including CONJ (Peng et al., 2024) and DRL (Zhang et al., 2024), to provide a more thorough evaluation.

## 5.2 Experimental Setup

In line with WOODS (Katz-Samuels et al., 2022), we employ a Wide ResNet architecture (Zagoruyko, 2016) with 40 layers and a width factor of 2 for the InD classifier $\phi_{\mathcal{S}_{in}}$. It is trained using stochastic gradient descent with a momentum of 0.9, weight decay of 0.0005, and an initial learning rate of 0.1. Training is performed for 100 epochs with cosine learning rate decay, a batch size of 128, and a dropout rate of 0.3. Hyperparameters $\epsilon$ and $k$ used in the proposed method Medix are selected from the sets {5e-5, 5e-4, 5e-3, 5e-2} and {4k, 7k, 10k, 20k}, respectively, taking into account dataset sizes and with the objective of maximizing OOD performance. For the OOD classifier $g_\theta$, we initialize it with the pre-trained InD classifier $\phi_{\mathcal{S}_{in}}$ and add a linear layer that performs binary classification using the penultimate-layer features. The learning rate for this classifier is set to 0.001, and fine-tuning is done for 100 epochs as outlined in Equation 6. We combine the binary classification loss with the InD classification loss, assigning a weight of 10 to the binary classification component. All other training parameters remain the same as those used for training $\phi_{\mathcal{S}_{in}}$.

Table 1: OOD detection performance comparison of Medix and baselines on CIFAR-10 as InD data. Performance averaged over five runs; best results are highlighted in **bold**.

| Methods | OOD Datasets | | | | | | | | | | | | InD ACC↑ |
|---|---|---|---|---|---|---|---|---|---|---|---|---|---|
| | SVHN | | PLACES365 | | LSUN-C | | LSUN-RESIZE | | TEXTURES | | Average | | |
| | FPR95↓ | AUROC↑ | FPR95↓ | AUROC↑ | FPR95↓ | AUROC↑ | FPR95↓ | AUROC↑ | FPR95↓ | AUROC↑ | FPR95↓ | AUROC↑ | |
| | | | | | Using $\mathbb{P}_{in}$ only | | | | | | | | |
| MSP | 48.49 | 91.89 | 59.48 | 88.20 | 30.80 | 95.65 | 52.15 | 91.37 | 59.28 | 88.50 | 50.04 | 91.12 | 94.84 |
| ODIN | 33.35 | 91.96 | 57.40 | 84.49 | 15.52 | 97.04 | 26.62 | 94.57 | 49.12 | 84.97 | 36.40 | 90.61 | 94.84 |
| Mahalanobis | 12.89 | 97.62 | 68.57 | 84.61 | 39.22 | 92.62 | 15.00 | 97.33 | 35.66 | 93.34 | 34.27 | 93.10 | 94.84 |
| Energy | 35.59 | 90.96 | 40.14 | 89.89 | 8.26 | 98.35 | 27.58 | 94.24 | 52.79 | 85.22 | 32.87 | 91.73 | 94.84 |
| KNN | 24.53 | 95.96 | 25.29 | 95.69 | 25.55 | 95.26 | 27.57 | 94.71 | 50.90 | 89.14 | 30.77 | 94.15 | 94.84 |
| ReAct | 40.76 | 89.57 | 41.44 | 90.44 | 14.38 | 97.21 | 33.63 | 93.58 | 53.63 | 86.59 | 36.77 | 91.48 | 94.84 |
| DICE | 35.44 | 89.65 | 46.83 | 86.69 | 6.32 | 98.68 | 28.93 | 93.56 | 53.62 | 82.20 | 34.23 | 90.16 | 94.84 |
| ASH | 6.51 | 98.65 | 48.45 | 88.34 | 0.90 | 99.73 | 4.96 | 98.92 | 24.34 | 95.09 | 17.03 | 96.15 | 94.84 |
| CSI | 17.30 | 97.40 | 34.95 | 93.64 | 1.95 | 99.55 | 12.15 | 98.01 | 20.45 | 95.93 | 17.36 | 96.91 | 94.17 |
| KNN+ | 2.99 | 99.41 | 24.69 | 94.84 | 2.95 | 99.39 | 11.22 | 97.98 | 9.65 | 98.37 | 10.30 | 97.99 | 93.19 |
| | | | | | Using $\mathbb{P}_{in}$ and $\mathbb{P}_{wild}$ | | | | | | | | |
| OE | 1.13 | 99.53 | 19.48 | 94.88 | 1.91 | 98.16 | 0.54 | 98.84 | 7.75 | 98.56 | 6.16 | 97.99 | 94.12 |
| Energy (w/ OE) | 5.24 | 98.72 | 14.66 | 96.18 | 2.35 | 99.30 | 4.85 | 98.62 | 10.51 | 97.10 | 7.52 | 97.98 | 94.24 |
| WOODS | 0.17 | 99.91 | 10.19 | 98.05 | 0.31 | 99.14 | 0.11 | 99.38 | 6.21 | 98.13 | 3.40 | 98.92 | 94.74 |
| Medix | **0.06** | **99.98** | **2.98** | **99.10** | **0.01** | **99.98** | **0.01** | **99.98** | **0.96** | **99.66** | **0.80** | **99.74** | 93.58 |
| (Ours) | ±0.01 | ±0.01 | ±0.29 | ±0.09 | ±0.01 | ±0.00 | ±0.01 | ±0.01 | ±0.13 | ±0.06 | ±0.09 | ±0.03 | ±0.64 |

Table 2: OOD detection performance comparison of Medix and baselines on CIFAR-100 as InD data. Performance averaged over five runs; best results are highlighted in **bold**.

| Methods | OOD Datasets | | | | | | | | | | | | InD ACC↑ |
|---|---|---|---|---|---|---|---|---|---|---|---|---|---|
| | SVHN | | PLACES365 | | LSUN-C | | LSUN-RESIZE | | TEXTURES | | Average | | |
| | FPR95↓ | AUROC↑ | FPR95↓ | AUROC↑ | FPR95↓ | AUROC↑ | FPR95↓ | AUROC↑ | FPR95↓ | AUROC↑ | FPR95↓ | AUROC↑ | |
| | | | | | Using $\mathbb{P}_{in}$ only | | | | | | | | |
| MSP | 84.59 | 71.44 | 82.84 | 73.78 | 66.54 | 83.79 | 82.42 | 75.38 | 83.29 | 73.34 | 79.94 | 75.55 | 75.96 |
| ODIN | 84.66 | 67.26 | 87.88 | 71.63 | 55.55 | 87.73 | 71.96 | 81.82 | 79.27 | 73.45 | 75.86 | 76.38 | 75.96 |
| Mahalanobis | 57.52 | 86.01 | 88.83 | 67.87 | 91.18 | 69.69 | 21.23 | 96.00 | 39.39 | 90.57 | 59.63 | 82.03 | 75.96 |
| Energy | 85.82 | 73.99 | 80.56 | 75.44 | 35.32 | 93.53 | 79.47 | 79.23 | 79.41 | 76.28 | 72.12 | 79.69 | 75.96 |
| KNN | 66.38 | 83.76 | 79.17 | 71.91 | 70.96 | 83.71 | 77.83 | 78.85 | 88.00 | 67.19 | 76.47 | 77.08 | 75.96 |
| ReAct | 74.33 | 88.04 | 81.33 | 74.32 | 39.30 | 91.19 | 79.86 | 73.69 | 67.38 | 82.80 | 68.44 | 82.01 | 75.96 |
| DICE | 88.35 | 72.58 | 81.61 | 75.07 | 26.77 | 94.74 | 80.21 | 78.50 | 76.29 | 76.07 | 70.65 | 79.39 | 75.96 |
| ASH | 21.36 | 94.28 | 68.37 | 71.22 | 15.27 | 95.65 | 68.18 | 85.42 | 40.87 | 92.29 | 42.81 | 87.77 | 75.96 |
| CSI | 64.70 | 84.97 | 82.25 | 73.63 | 38.10 | 92.52 | 91.55 | 63.42 | 74.70 | 92.66 | 70.26 | 81.44 | 69.90 |
| KNN+ | 32.21 | 93.74 | 68.30 | 75.31 | 40.37 | 86.13 | 44.86 | 88.88 | 46.26 | 87.40 | 46.40 | 86.29 | 73.78 |
| | | | | | Using $\mathbb{P}_{in}$ and $\mathbb{P}_{wild}$ | | | | | | | | |
| OE | 2.86 | 99.05 | 40.21 | 88.75 | 4.13 | 99.05 | 1.25 | 99.38 | 22.86 | 94.63 | 14.26 | 96.17 | 73.38 |
| Energy (w/ OE) | 2.71 | 99.34 | 34.82 | 90.05 | 3.27 | 99.18 | 2.54 | 99.23 | 30.16 | 94.76 | 14.70 | 96.51 | 72.76 |
| WOODS | 0.17 | 99.80 | 21.87 | 93.73 | 0.48 | 99.61 | 1.24 | 99.54 | 9.95 | 95.97 | 6.74 | 97.73 | 73.91 |
| Medix | **0.16** | **99.96** | **15.99** | **95.23** | **0.13** | **99.98** | **0.83** | **99.83** | **8.02** | **97.79** | **5.42** | **98.96** | 73.33 |
| (Ours) | ±0.02 | ±0.00 | ±0.66 | ±0.14 | ±0.06 | ±0.02 | ±0.36 | ±0.06 | ±0.75 | ±0.30 | ±0.37 | ±0.10 | ±0.83 |

## 5.3 Results

We present our main results in Table 2 on CIFAR-100, where Medix outperforms all OOD detection baselines. The results highlight the following key observations: (1) Methods trained on both InD and wild data significantly outperform those trained exclusively on InD data. Medix reduces the FPR95 by 52.31% on PLACES365 and 38.24% on TEXTURES compared to KNN+, demonstrating the effectiveness of incorporating in-the-wild data for model regularization. (2) Medix further outperforms competitive methods utilizing wild data ($\mathbb{P}_{wild}$): On CIFAR-100, Medix achieves an average FPR95 of 5.42%, which represents a 1.32%

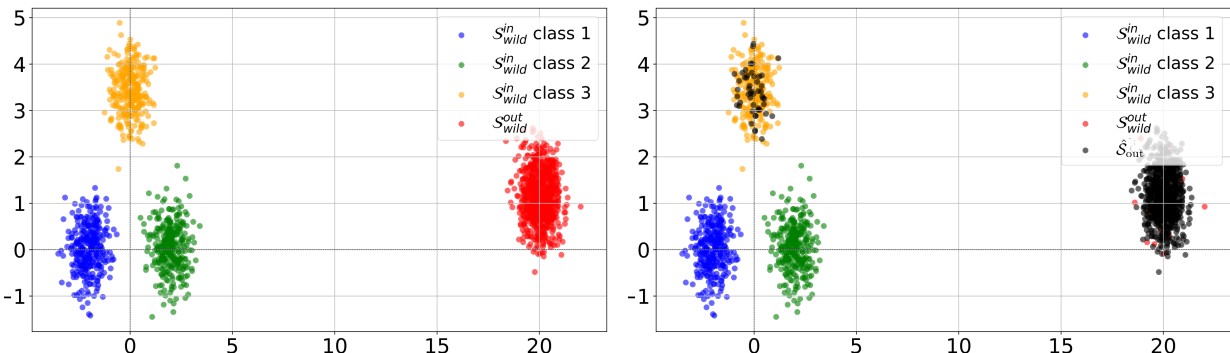

Figure 3: Example of Medix applied to unlabeled wild data. (a) Setup of the InD data $\mathcal{S}_{\text{wild}}^{\text{in}}$ and OOD data $\mathcal{S}_{\text{wild}}^{\text{out}}$ in the wild, with inliers sampled from three multivariate Gaussian distributions. (b) Outliers $\hat{\mathcal{S}}_{\text{out}}$ filtered by Medix (in black), with an error rate of $\hat{\mathcal{S}}_{\text{out}}$ containing InD data $\mathcal{S}_{\text{wild}}^{\text{in}}$ is only 12.5%.

improvement over WOODS. Additionally, Medix maintains a competitive InD accuracy of 73.33%. This slight difference can be attributed to the fact that our method is trained on 25,000 labeled InD samples, while baseline methods, which do not leverage wild data, use the full CIFAR-100 training set of 50,000 samples. We present the results for CIFAR-10 in Table 1, where Medix surpasses all baseline methods. Medix outperforms WOODS by 7.21% on PLACES365 and 5.25% on TEXTURES in terms of FPR95, demonstrating its effectiveness in detecting OOD samples.

**A representative visual example of Medix.** We further investigate the performance of Algorithm 1 (Outlier Extraction) in extracting OOD samples from wild data $\mathcal{S}_{\text{wild}}$. To visualize this, we design an experiment using 2-dimensional synthetic data. This simulation is designed to be simple to facilitate better understanding. We generate the InD data by sampling from three multivariate Gaussian distributions, corresponding to three classes. The mean vectors are set to $[-2, 0]$, $[2, 0]$, and $[0, 2\sqrt{3}]$, respectively. The covariance matrix for all three classes is fixed at $0.25 \cdot \mathcal{I}$. For the OOD data, we use a multivariate Gaussian distribution $\mathcal{N}([20, 2\sqrt{3}], 0.25 \cdot \mathcal{I})$. In Figure 3, we observe that our method successfully identifies outliers, the majority of which align with the ground truth. Medix successfully flags 87.5% of actual OOD samples as outliers, underscoring its robustness in outlier extraction.

**Additional Experiments.** We defer additional experiments to Appendix A, which include (1) ablation studies on hyperparameter selection and sensitivity analysis (Appendix A.2), showing Medix's strong robustness to hyperparameters, (2) a comparison between EWM and geometric median (Appendix A.1), showing that EWM is more sensitive to distributional shifts, making it more effective and reliable choice for filtering in our method, (3) a comprehensive comparison with methods employing competitive contrastive learning objectives (Appendix A.3), demonstrating that Medix outperforms even these competitive baselines by a significant margin, (4) evaluation on large-scale data under the complex unseen OOD setting, where $\mathbb{P}_{\text{out}}^{\text{test}} \neq \mathbb{P}_{\text{out}}$ (Appendix A.4), demonstrating that Medix outperforms baselines by a significant margin, (5) evaluating computation and memory efficiency of Medix (Appendix A.6) (6) evaluating the impact of pseudo-label quality (Appendix A.5), showing that our method is resilient to noisy or low-confidence labels, and (7) a comparison with semi-supervised open-set recognition methods (Appendix A.7), (8) a scalability analysis of Medix across larger architectures, showing that the L2-based filtering signal remains robust even in high-dimensional gradient spaces (Appendix A.8), (9) evaluation on ImageNet-1k, demonstrating Medix's strong generalization to high-resolution, large-scale datasets (Appendix A.9), and (10) a comparison against a naive two-stage baseline that replaces our median filter with MSP-filtering (Appendix A.10), confirming that Medix's gains stem from the precision of its median filtering rather than the two-stage pipeline alone.

# 6 Related Work

In recent years, there has been a growing interest in OOD detection (Fort et al., 2021; Yang et al., 2024; Fang et al., 2022; Zhu et al., 2022; Yang et al., 2022; Wang et al., 2022c; Galil et al., 2023; Djurisic et al.,

2023; Tao et al., 2023; Zheng et al., 2023; Wang et al., 2022b; 2023b; Uppaal et al., 2023; Zhu et al., 2023; Bai et al., 2023; Ming & Li, 2024; Zhang et al., 2023; Ghosal et al., 2024; Abbas et al., 2025). One approach to detect OOD data uses scoring functions to assess data distribution, including distance-based methods (Lee et al., 2018a; Tack et al., 2020a; Ren et al., 2021; Sehwag et al., 2021; Sun et al., 2022; Du et al., 2022a; Ming et al., 2023; Ren et al., 2022), gradient-based score (Huang et al., 2021), energy-based score (Liu et al., 2020b; Wang et al., 2021; Wu et al., 2023), confidence-based approaches (Bendale & Boult, 2016; Hendrycks & Gimpel, 2017; Liang et al., 2018), and Bayesian methods (Gal & Ghahramani, 2016; Lakshminarayanan et al., 2017; Maddox et al., 2019; Malinin & Gales, 2019; Wen et al., 2020; Kristiadi et al., 2020).

Another approach to OOD detection involves using regularization techniques during the training phase (Malinin & Gales, 2018; Geifman & El-Yaniv, 2019; Hein et al., 2019; Meinke & Hein, 2020; Jeong & Kim, 2020; Liu et al., 2020a; Van Amersfoort et al., 2020; Yang et al., 2021; Wei et al., 2022; Du et al., 2022b; 2023; Wang et al., 2023a). For example, regularization techniques can be applied to the model to either reduce its confidence (Lee et al., 2017; Hendrycks et al., 2019) or increase its energy (Liu et al., 2020b; Du et al., 2022c; Ming et al., 2022) on the OOD data. Most of these regularization methods assume the availability of an auxiliary OOD dataset.

Several studies (Zhou et al., 2021; Katz-Samuels et al., 2022; He et al., 2023) have relaxed the assumption of using only labeled data by incorporating unlabeled wild data (Katz-Samuels et al., 2022; Geng et al., 2025), though they did not propose a clear mechanism for outlier detection. In contrast, Du et al. (2024a;b) introduced an explicit outlier filtering method, but their thresholding technique differs fundamentally from ours, as we utilize a new median-centric approach to detect the outliers. Additionally, Katz-Samuels et al. (2022); Du et al. (2024a) operate under the assumption of batch-level mixing, where each batch has a set ratio of InD and OOD samples. However, with large outsourced datasets, such structured mixing is not available-data is mixed randomly across the dataset. Our method addresses this by enabling dataset-level mixing without relying on batch-level structure. Many studies also leverage positive-unlabeled learning, which trains classifiers using positive and/or unlabeled data (Letouzey et al., 2000; Hsieh et al., 2015; Plessis et al., 2015; Niu et al., 2016; Gong et al., 2018; Chapel et al., 2020; Garg et al., 2021; Xu & Denil, 2021; Garg et al., 2022; Zhao et al., 2022; Acharya et al., 2022). However, a key distinction from our approach is that these methods focus solely on differentiating $\mathbb{P}_{out}$ from $\mathbb{P}_{in}$, without simultaneously training an OOD classifier. Additionally, we propose a median-centric method to identify outliers in unlabeled data, with provably low error rates.

Generation-based OOD methods (Marek et al., 2021; Yoon et al., 2023; Gao et al., 2025; Abbas et al., 2025; Yoon et al., 2025; Lee et al., 2025) represent an important complementary line of research. These approaches synthesise OOD samples using diffusion models, GANs, normalising flows, or LLMs, and then train OOD detectors on the synthetic data. Unlike Medix, which extracts outliers from real unlabeled wild data, generation-based methods do not require a wild mixture but instead rely on a generative model to produce plausible OOD proxies. While Medix offers theoretical guarantees on extraction error and computational simplicity, generation-based methods can be advantageous when no wild data is available or when large-scale synthetic augmentation is desired. A comprehensive comparison of these two paradigms is an interesting direction for future work.

## 7 Conclusions

In this work, we introduced Medix, a novel median-centric framework for OOD detection that leverages unlabeled in-the-wild data. Using the inherent robustness of median operation, Medix effectively filters outliers from mixed unlabeled data, enabling the training of a reliable OOD detector. Our theoretical analysis established provable bounds on the inlier misclassification rate, demonstrating that Medix maintains robustness even under significant OOD contamination (up to 50%), with errors controlled by sub-Gaussian concentration and contamination effects. We also provided complementary theoretical limits on the rate of OOD misclassification to accurately isolate OOD samples under clear separation conditions. Empirical validation across diverse benchmarks showcased Medix's performance over 20 baselines: it reduced the average FPR95 by 40.98% compared to strong baselines like KNN+ and outperformed state-of-the-art methods such as WOODS and DRL, while achieving an outlier extraction error rate as low as 12.5%.

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

# A  Additional Experiments

## A.1  Element-wise Median vs Geometric Median

Another pertinent question to ask is why we chose the element-wise median over the geometric median (Acharya et al., 2024) for outlier detection in Medix. To answer this question, we conducted a preliminary experiment using CIFAR-10 as the InD dataset and SVHN as the OOD dataset. Specifically, $\mathcal{S}_{\text{wild}}$ consists of 5k samples drawn from CIFAR-10, ensuring that these samples are disjoint from the training set used to train the model $\phi_{\mathcal{S}_{\text{in}}}$, which we leverage to compute $\bar{\nabla}_{\text{in}}$. We incrementally add SVHN OOD samples to $\mathcal{S}_{\text{wild}}$ and track the behavior of the $L_2$-norm deviation between $\bar{\nabla}_{\text{in}}$ and the element-wise median of the gradients of the wild dataset as well as the proportion of OOD samples removed. The results, shown in the Figure 4, demonstrate that for the same number of OOD samples in the wild dataset, the element-wise median identified a significantly higher proportion of OOD samples as outliers compared to the geometric median. This indicates that the element-wise median is more sensitive to distributional shifts, making it a more effective and reliable choice for filtering in our method.

Table 3: Effect of hyperparameters $\epsilon$ and $k$ on OOD detection.

| Method | FPR95↓ | $\epsilon$ | $k$ |
|---|---|---|---|
| DICE | 88.35 | – | – |
| ASH | 21.36 | – | – |
| CSI | 64.70 | – | – |
| KNN+ | 32.21 | – | – |
| OE | 2.86 | – | – |
| Energy | 2.71 | – | – |
| Medix | 0.16 | 0.005 | 20000 |
| Medix | 0.20 | 0.0005 | 20000 |
| Medix | 0.68 | 0.005 | 10000 |

## A.2  Hyperparameter Selection and Sensitivity Analysis

We conducted an ablation study to assess the sensitivity of Medix to the hyperparameters $\epsilon$ and $k$. For this experiment, we employed CIFAR-100 as the InD dataset and SVHN as the OOD dataset. As shown in Table 3, Medix exhibits strong robustness to variations in the values of $\epsilon$ and $k$. In particular, Medix achieves a low FPR95 of 0.16 when using $\epsilon = 0.005$ and $k = 20000$. Even when the values of $\epsilon$ and $k$ are varied—e.g., reducing $\epsilon$ to 0.0005 or halving $k$ to 10000—the performance remains competitive (FPR95 of 0.20 and 0.68, respectively), demonstrating a graceful degradation rather than a sharp drop. The results indicate that while optimal hyperparameter selection is important, Medix maintains its effectiveness across a variety of hyperparameter choices, surpassing the baselines. This insensitivity to exact hyperparameter tuning makes Medix a reliable choice in real-world deployments, where exhaustive tuning may not be feasible.

## A.3  Comprehensive comparison with recent competitive methods

We have extended our comparisons to include competitive baselines that employ diverse contrasting learning objectives, such as SSD+ (Sehwag et al., 2021), ProxyAnchor (Kim et al., 2020), and CIDER Ming et al. (2023). Although these methods use contrasting learning objectives, which we do not, we included them for a comprehensive comparison. Additionally, we included more recent methods, such as CONJ (Peng et al.,

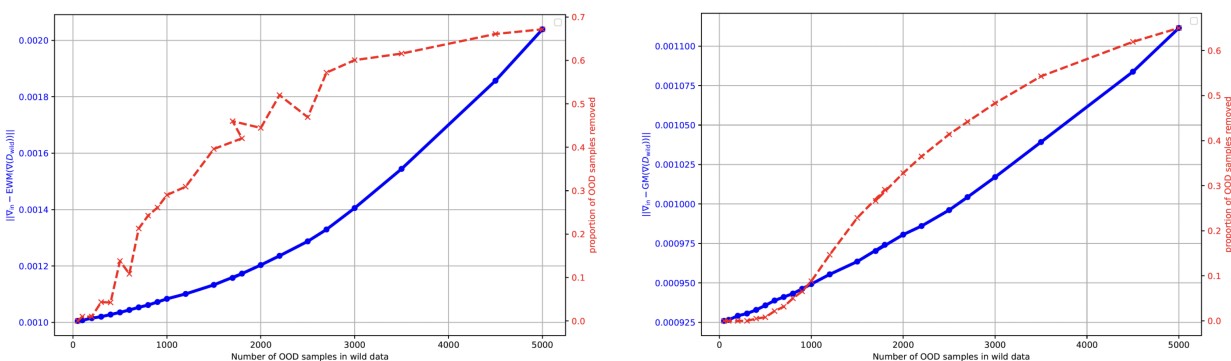

Figure 4: Comparison of element-wise median (EWM) and geometric median (GM).

Table 4: OOD detection performance comparison of Medix and recent competitive baselines on CIFAR-100 as InD data. Performance averaged over five runs; best results are highlighted in **bold**.

| Methods | OOD Datasets | | | | | | | | | |
| | SVHN | | PLACES365 | | LSUN-C | | TEXTURES | | AVERAGE | |
| | FPR95↓ | AUROC↑ | FPR95↓ | AUROC↑ | FPR95↓ | AUROC↑ | FPR95↓ | AUROC↑ | FPR95↓ | AUROC↑ |
|---|---|---|---|---|---|---|---|---|---|---|
| MSP | 78.89 | 79.80 | 84.38 | 74.21 | 83.47 | 75.28 | 86.51 | 72.53 | 83.31 | 75.46 |
| Mahalanobis | 87.09 | 80.62 | 84.63 | 73.89 | 84.15 | 79.43 | 61.72 | 84.87 | 79.40 | 79.70 |
| ODIN | 70.16 | 84.88 | 82.16 | 75.19 | 76.36 | 80.10 | 85.28 | 75.23 | 78.49 | 78.85 |
| Energy | 66.91 | 85.25 | 81.41 | 76.37 | 59.77 | 86.69 | 79.01 | 79.96 | 71.78 | 82.07 |
| ReAct | 50.93 | 88.75 | 83.55 | 73.10 | 64.02 | 80.31 | 64.40 | 81.95 | 65.72 | 81.03 |
| KNN | 46.25 | 90.39 | 82.08 | 75.44 | 60.85 | 85.61 | 62.39 | 83.95 | 62.89 | 83.85 |
| Vim | 73.42 | 84.62 | 85.34 | 69.34 | 86.96 | 69.74 | 74.56 | 76.23 | 80.07 | 74.98 |
| VOS | 43.24 | 82.80 | 76.85 | 78.63 | 73.61 | 84.69 | 57.57 | 87.31 | 62.82 | 83.36 |
| CSI | 44.53 | 92.65 | 79.08 | 76.27 | 75.58 | 83.78 | 61.61 | 86.47 | 65.20 | 84.79 |
| ProxyAnchor | 87.21 | 82.43 | 70.10 | 79.84 | 37.19 | 91.68 | 65.64 | 84.99 | 65.04 | 84.74 |
| SSD+ | 31.19 | 94.19 | 77.74 | 79.90 | 79.39 | 85.18 | 66.63 | 86.18 | 63.74 | 86.36 |
| KNN+ | 39.23 | 92.78 | 80.74 | 77.58 | 48.99 | 89.30 | 57.15 | 88.35 | 56.53 | 87.00 |
| ASH | 52.96 | 90.19 | 72.62 | 76.38 | 75.18 | 76.52 | 56.17 | 86.75 | 64.23 | 82.46 |
| CIDER | 23.09 | 95.16 | 79.63 | 73.43 | 16.16 | 96.33 | 43.87 | 90.42 | 40.69 | 88.84 |
| CONJ | 46.19 | 90.44 | 80.81 | 75.83 | 60.45 | 85.90 | 62.13 | 83.77 | 62.40 | 83.99 |
| DRL | 20.15 | 94.07 | 76.64 | 77.55 | 16.97 | 94.63 | 31.97 | 92.09 | 36.43 | 89.59 |
| Medix | **0.48** | **99.87** | **24.52** | **93.42** | **0.73** | **99.84** | **8.99** | **97.92** | **8.68** | **97.76** |
| (Ours) | $\pm 0.05$ | $\pm 0.02$ | $\pm 1.76$ | $\pm 0.36$ | $\pm 0.08$ | $\pm 0.03$ | $\pm 0.72$ | $\pm 0.24$ | $\pm 0.65$ | $\pm 0.16$ |

2024), DRL (Zhang et al., 2024), Vim (Wang et al., 2022a), and VOS (Du et al., 2022c), to provide a more thorough evaluation. For this experiment, we use CIFAR-100 as InD dataset and train the ResNet-34 model[1] following the setup in Ming et al. (2023); Zhang et al. (2024). The model is trained using stochastic gradient descent with momentum 0.9, and weight decay $10^{-4}$ for 500 epochs. The initial learning rate is 0.5 with cosine scheduling and the batch size is 512. We use the checkpoints provided by Ming et al. (2023)[2]. The results, as shown in Table 4, demonstrate that Medix outperforms even these competitive baselines by a significant margin, including those trained with contrastive learning objectives, achieving an average FPR95 of 8.68% and an average AUROC of 97.76%.

## A.4 Evaluation on Large-Scale, Complex Unseen OOD Settings

We next investigate whether Medix can handle large-scale wild OOD data under the unseen OOD setting. This setting is more challenging for two main reasons: (1) it leverages a large-scale wild OOD dataset ($\mathbb{P}_{out}$), increasing both data complexity and computational demand, and (2) the test-time OOD distribution ($\mathbb{P}_{out}^{test}$) is intentionally chosen to be distributionally different from the wild OOD data used during training, i.e., $\mathbb{P}_{out}^{test} \neq \mathbb{P}_{out}$, which better reflects realistic and challenging deployment scenarios. We used CIFAR-100 as the labeled InD data, 300K Random Images dataset Hendrycks et al. (2019) as the unlabeled wild OOD data and tested on the SVHN dataset as the test OOD data. This setup introduces a greater level of complexity because the large-scale wild OOD data (300K Random Images) is significantly different from the OOD test data (SVHN). To evaluate the performance of our approach, we compare it against baselines that also leverage wild data in training. The results in the

Table 5: Comparison of OOD detection performance on large-scale unseen OOD data.

| Method | FPR95↓ | AUROC↑ |
|---|---|---|
| OE | $68.80 \pm 2.8$ | $82.89 \pm 1.1$ |
| Energy (w/ OE) | $69.81 \pm 2.4$ | $85.59 \pm 1.0$ |
| WOODS | $69.41 \pm 2.7$ | $86.76 \pm 0.8$ |
| Medix (ours) | $\mathbf{41.29 \pm 1.2}$ | $\mathbf{87.25 \pm 0.9}$ |

Table 5 highlight the superior performance of our method, Medix, compared to the baselines, with a significantly lower FPR95 ($41.29 \pm 1.2$) and higher AUROC ($87.25 \pm 0.6$), showing its effectiveness in distinguishing between InD and OOD data.

---

[1]Since ResNet-34 is the standard model across these works, we adopt the same model to ensure consistency and facilitate a fair comparison.

[2]https://github.com/deeplearning-wisc/cider

### A.5 Evaluating the Impact of Pseudo-Label Quality on OOD Filtering and Robustness

Another important question to ask is whether the quality of pseudo-labels $\hat{y}_{\tilde{x}_i}$, particularly in terms of softmax confidence, affects the performance of OOD filtering, and whether a simple pre-filtering step that removes low-confidence pseudo-labels could improve the robustness of the model. To answer this question, we conducted an experiment to evaluate how filtering low-confidence pseudo-labels affects OOD filtering. For this experiment, we used CIFAR-10 as the InD dataset and LSUN-Resize as the OOD dataset. Specifically, we computed the softmax probabilities for each pseudo-labeled sample and discarded those with low-confidence predictions, setting a threshold of 0.6. After filtering, we found that 15.98% of the samples were removed from the training set.

The results showed that there was virtually no difference in performance between the method with filtering and the method without filtering. Both methods yielded a FPR95 of 0.01%, but with filtering, AUROC increased slightly from 99.98% to 99.99%. Thus, removing low-confidence pseudo-labels doesn't significantly impact the robustness of the model or its ability to detect OOD samples, showing that our method is resilient to noisy or low-confidence labels, and further filtering steps are unlikely to yield meaningful improvements.

### A.6 Evaluating Computation and Memory Efficiency in Medix Filtering

We now turn to the question of whether Medix's filtering phase is computationally feasible and memory-efficient on large datasets. We conducted profiling experiments on an NVIDIA A100-SXM4-80GB GPU using approximately 15,000 unlabeled samples for each InD–OOD pair. For this experiment, we used CIFAR-10 and CIFAR-100 as the InD datasets and LSUN-Resize as the OOD dataset. As seen from the results in Table 6, the GPU memory usage remains modest, and the filtering stage completes in roughly 75 to 91 minutes on an A100 for 15k samples. This demonstrates that the Medix filtering process is computationally feasible and does not impose excessive memory overhead, even on large datasets.

Table 6: Profiling results for Medix filtering on NVIDIA A100 80GB GPU.

| InD–OOD | Wall Clock Time (s) | Peak GPU Memory (MB) | Current GPU Memory (MB) |
|---|---|---|---|
| CIFAR10 – LSUN-Resize | 4497.17 | 99.46 | 31.74 |
| CIFAR100 – LSUN-Resize | 5478.36 | 99.56 | 31.79 |

### A.7 Comparison with Semi-Supervised Open-Set Recognition Methods

Our work differs from recent semi-supervised open-set recognition methods (Saito et al., 2021; Fan et al., 2023; Hang & Zhang, 2024; Wang et al., 2024) in several key ways. For example, in OpenMatch Saito et al. (2021), the main problem is to handle outliers in semi-supervised learning (SSL) when training a standard classifier, whereas in our setting, the main challenge is to detect OOD samples from unlabeled wild data and train a dedicated OOD detector classifier. In other words, while SSL methods (Saito et al., 2021; Fan et al., 2023; Hang & Zhang, 2024; Wang et al., 2024) aim to improve classification despite outliers, our approach enhances OOD detection in an open-world setting where labeled OOD data is unavailable.

Secondly, these SSL methods aim to train a classifier that is robust to OOD samples in SSL, treating outliers as noise, while in our setting, the goal is to explicitly detect OOD samples and train an OOD detector. In other words, SSL methods focus on suppressing OOD samples during training to improve SSL performance, whereas Medix actively detects and leverages these OOD samples to build a more effective and reliable OOD detection system.

Lastly, the two approaches differ in their techniques: OpenMatch is based on consistency regularization, where the main idea is to enforce consistency across different stochastic transformations of the same input. This is mathematically modeled through soft constraints on the classifier's outputs, encouraging smooth decision boundaries. On the other hand, Medix relies on median-based filtering for outlier detection. The method uses statistical robustness properties of the median, leveraging its insensitivity to extreme values to identify potential OOD samples. The theoretical analysis derives error bounds based on the contamination

effect (proportion of OOD samples) and the concentration effect (sub-Gaussian behavior of InD gradients), ensuring that OOD detection error remains low.

### A.8 Scalability Analysis of Medix Across Larger Architectures

We further evaluate Medix on a larger backbone to understand its behavior in high-dimensional gradient spaces. Specifically, we use ResNet-50, where the penultimate layer gradient dimensionality is $d = 2048$, significantly higher than in the architectures used in the main experiments (e.g. WideResNet). We follow the same experimental setup as in Table 2. CIFAR-100 is used as the InD dataset, and we evaluate on two challenging OOD datasets: TEXTURES and PLACES. We also implement WOODS (Katz-Samuels et al., 2022) on ResNet-50 for a direct comparison.

As shown in Table 7, Medix consistently outperforms WOODS by a large margin across both OOD datasets. Notably, Medix achieves an FPR95 of 1.36 on TEXTURES and 4.13 on PLACES, while maintaining AUROC scores above 98% in both cases.

Despite the substantially higher dimensionality of the gradient space ($d = 2048$), the L2 distance between the mean InD gradient and the EWM of wild gradients remains highly discriminative. In fact, performance is on par with or better than results obtained using smaller architectures (c.f. Table 1-2).These findings indicate that Medix scales well to larger models such as ResNet-50, and that the L2-based filtering mechanism remains robust even in high-dimensional settings.

Table 7: Performance comparison on ResNet-50 with CIFAR-100 as InD, and TEXTURES and PLACES as OOD datasets.

| Method | OOD | FPR95 ↓ | AUROC ↑ |
|--------|-----|---------|---------|
| WOODS | TEXTURES | 11.93 | 97.56 |
| Medix | TEXTURES | **1.36** | **99.66** |
| WOODS | PLACES | 47.85 | 88.69 |
| Medix | PLACES | **4.13** | **98.82** |

### A.9 Evaluation on ImageNet-1k and Large-Scale OOD Detection

To assess Medix's generalization to high-resolution, large-scale datasets, we conduct experiments on ImageNet-1k using ResNet-50, with ImageNet-1k as the InD dataset. We evaluate on two OOD datasets of differing difficulty: SVHN, a well-separated OOD source, and TEXTURES, a more challenging one. This setup tests Medix's ability to generalize to high-resolution images ($224{\times}224$) and a substantially larger number of classes than CIFAR-100. We additionally evaluate WOODS (Katz-Samuels et al., 2022) under identical conditions, adapting its open-source implementation to the ResNet-50 architecture[3].

As shown in Table 8, on the well-separated SVHN benchmark, Medix achieves near-perfect performance (FPR95 of 0.00, AUROC of 100), with an ID-vs-OOD accuracy of 98.98%; WOODS performs comparably (FPR95 of 0.01, AUROC of 99.98). Because both methods saturate on this pair, we additionally report results on TEXTURES, a harder OOD dataset where the two methods are more clearly differenti-

Table 8: OOD detection performance with ImageNet-1k as InD, using ResNet-50.

| Method | OOD | FPR95 ↓ | AUROC ↑ |
|--------|-----|---------|---------|
| WOODS | SVHN | 0.01 | 99.98 |
| Medix | SVHN | **0.00** | **100.00** |
| WOODS | TEXTURES | **5.43** | **98.89** |
| Medix | TEXTURES | 7.53 | 97.98 |

ated. Here, both methods remain strong: WOODS attains an FPR95 of 5.43 and AUROC of 98.89, while Medix attains an FPR95 of 7.53 and AUROC of 97.98. Although WOODS is slightly ahead on this particular benchmark, both methods maintain low false-positive rates and high AUROC at ImageNet scale, confirming that Medix's gradient-median filtering signal remains effective and competitive in high-resolution, large-class settings beyond the CIFAR benchmarks used in our main experiments.

### A.10 Comparison with a Naive Two-Stage Filtering Baseline

A natural question is whether the gains of Medix stem from its two-stage architecture in general, or specifically from the median-based filtering used in Stage 1. To isolate this, we construct a baseline that retains the exact Stage 2 binary-detector training of Medix but replaces our median filter with a simple, off-the-shelf filter: we score each wild sample with Maximum Softmax Probability (MSP) (Hendrycks & Gimpel, 2017) from the

---

[3]for WOODS, we used their open-source code: https://github.com/jkatzsam/woods_ood

InD-trained model, flag the low-confidence samples as candidate outliers, and train the same binary OOD detector on the resulting set together with the labeled InD data. We refer to this as *MSP-filter + binary*. We evaluate on SVHN and TEXTURES as OOD datasets, using CIFAR-10 and CIFAR-100 as InD, and report raw (InD-only) MSP and competitive wild-data baselines for context.

Table 9: Isolating the role of median filtering. We compare Medix against a naive two-stage baseline (*MSP-filter + binary*) that uses the same Stage-2 binary detector but replaces our median filter with MSP-based filtering. Results on SVHN and TEXTURES as OOD, with CIFAR-10 and CIFAR-100 as InD. Raw InD-only MSP and wild-data baselines are shown for reference. Best results in **bold**.

| Methods | CIFAR-10 (InD) | | | | CIFAR-100 (InD) | | | |
| | SVHN | | TEXTURES | | SVHN | | TEXTURES | |
| | FPR95↓ | AUROC↑ | FPR95↓ | AUROC↑ | FPR95↓ | AUROC↑ | FPR95↓ | AUROC↑ |
| | Using $\mathbb{P}_{\text{in}}$ only | | | | | | | |
| MSP | 48.49 | 91.89 | 59.28 | 88.50 | 84.59 | 71.44 | 83.29 | 73.34 |
| | Using $\mathbb{P}_{\text{in}}$ and $\mathbb{P}_{\text{wild}}$ | | | | | | | |
| OE | 1.13 | 99.53 | 7.75 | 98.56 | 2.86 | 99.05 | 22.86 | 94.63 |
| Energy (w/ OE) | 5.24 | 98.72 | 10.51 | 97.10 | 2.71 | 99.34 | 30.16 | 94.76 |
| WOODS | 0.17 | 99.91 | 6.21 | 98.13 | 0.17 | 99.80 | 9.95 | 95.97 |
| MSP-filter + binary | 24.18 | 95.32 | 52.62 | 88.83 | 35.43 | 92.12 | 55.54 | 88.86 |
| Medix (Ours) | **0.06**±0.01 | **99.98**±0.01 | **0.96**±0.13 | **99.66**±0.06 | **0.16**±0.02 | **99.96**±0.00 | **8.02**±0.75 | **97.79**±0.30 |

As shown in Table 9, the *MSP-filter + binary* baseline does improve over the raw InD-only MSP detector in several cases (e.g., FPR95 drops from 84.59 to 35.43 on CIFAR-100/SVHN, and from 48.49 to 24.18 on CIFAR-10/SVHN), confirming that the Stage-2 binary detector recovers some signal. However, it remains dramatically worse than Medix across all four settings: Medix reduces FPR95 from 35.43 to 0.16 on CIFAR-100/SVHN and from 24.18 to 0.06 on CIFAR-10/SVHN, with similarly large margins on TEXTURES. The naive baseline also trails the existing wild-data methods (OE, Energy w/ OE, WOODS), placing it well below the rest of the $\mathbb{P}_{\text{in}} + \mathbb{P}_{\text{wild}}$ group.

This gap is explained directly by filtering precision. Because MSP separates InD from OOD poorly (its own FPR95 exceeds 80% on CIFAR-100/SVHN), the candidate outlier set it produces is heavily contaminated with InD samples. Training the Stage-2 detector on this noisy set propagates the contamination, yielding a weak final detector. In contrast, Medix extracts outliers with only a 12.5% error rate (Figure 3), and this cleaner set enables the same Stage-2 detector to reach near-perfect performance. The effect is most pronounced on the harder TEXTURES, where MSP filtering barely improves over the raw detector (e.g., CIFAR-10/TEXTURES: FPR95 from 59.28 → 52.62, and AUROC from 88.50 → 88.83), indicating that contamination dominates when the underlying scoring function is unreliable. Together, these results confirm that Medix's advantage is driven by the quality of its median-based filtering, not merely by the two-stage pipeline: replacing the median filter with a naive alternative in the identical pipeline causes performance to collapse.

### A.11 Details on Statistical Operations for Outlier Removal

To avoid ambiguity, we explicitly define the two key operations used in our outlier extraction procedure.

**Mean** $(\bar{\nabla}_{\text{in}})$. Let $\{g_i\}_{i=1}^n$ be the set of gradient vectors computed from the labeled InD training set, where each $g_i \in \mathbb{R}^d$ (with $d$ the number of parameters of the chosen layer). The mean gradient $\bar{\nabla}_{\text{in}}$ is the $d$-dimensional vector whose $j$-th coordinate is the average of the $j$-th coordinates of all $g_i$:

$$(\bar{\nabla}_{\text{in}})_j = \frac{1}{n} \sum_{i=1}^n g_{i,j}, \qquad j = 1, \ldots, d,$$

where $g_{i,j}$ denotes the $j$-th coordinate of $g_i$. In words, we take the element-wise average across the sample dimension.

**Element-wise median (EWM).** For a subset $S \subseteq S_{\text{wild}}$ with $|S| = m$, we arrange the corresponding gradient vectors into a matrix $G_S$ of size $m \times d$, where row $i$ corresponds to the gradient of the $i$-th sample

in $S$. Then $\text{EWM}(G_S)$ is the $d$-dimensional vector obtained by taking the median across rows (i.e., over the samples) for each coordinate independently:

$$\big[\text{EWM}(G_S)\big]_j \;=\; \text{median}\big(\{\,[G_S]_{i,j} \mid i = 1, \ldots, m\,\}\big), \qquad j = 1, \ldots, d.$$

Thus, for each parameter coordinate, we compute the median of the values observed across all samples in $S$. This operation is robust to outliers and serves as the core statistic for our greedy removal algorithm (Algorithm 1).

These definitions are consistent with the implementation (e.g., `torch.mean(dim=0)` and `torch.median(dim=0)` in PyTorch, where `dim=0` indicates the sample axis).

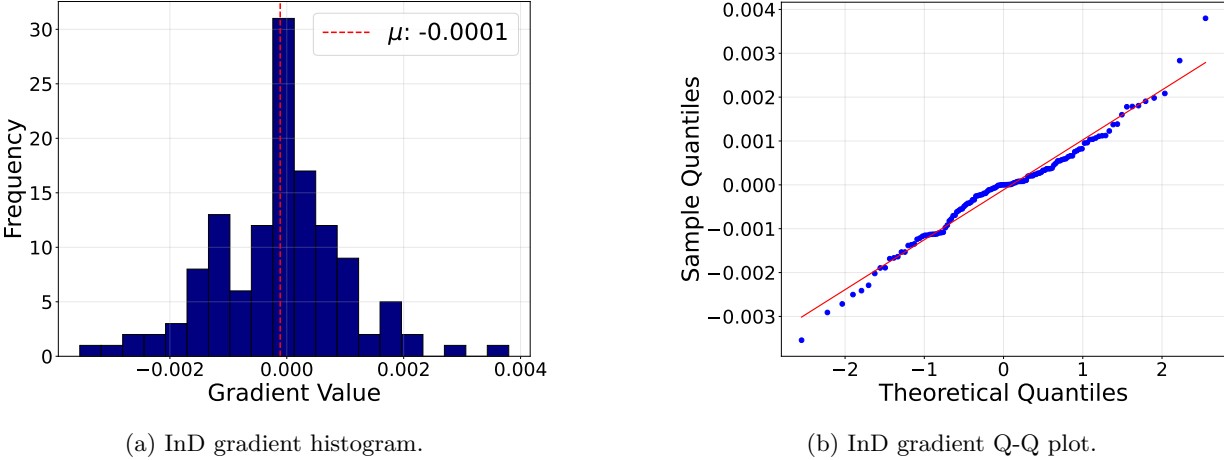

(a) InD gradient histogram.  (b) InD gradient Q-Q plot.

Figure 5: Illustration of InD sample gradients exhibiting sub-Gaussian behavior in each coordinate. (left) Histogram of gradient values (CIFAR-100 InD data) showing concentration around the mean with light tails, consistent with sub-Gaussianity. (right) Q-Q plot comparing empirical quantiles of InD gradients against a theoretical Gaussian distribution, confirming alignment with sub-Gaussianity.

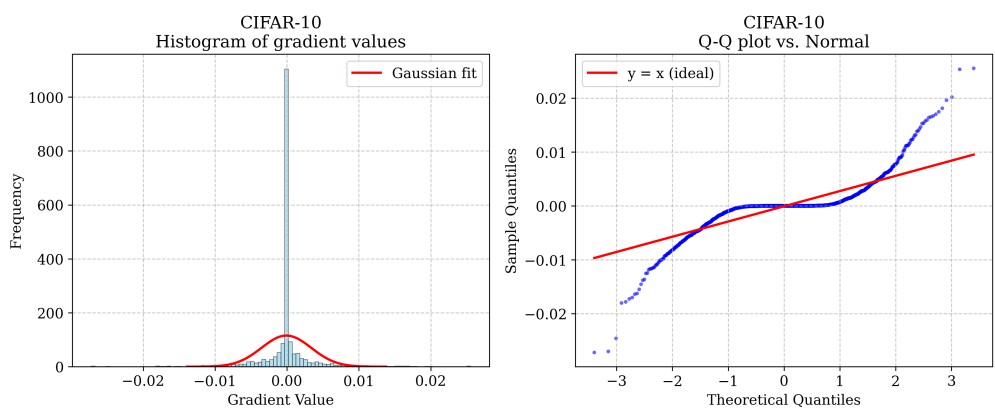

Figure 6: Illustration of InD sample gradients exhibiting sub-Gaussian behavior in each coordinate. (left) Histogram of gradient values (CIFAR-10 InD data) showing concentration around the mean with light tails, consistent with sub-Gaussianity. (right) Q-Q plot comparing empirical quantiles of InD gradients against a theoretical Gaussian distribution, confirming alignment with sub-Gaussianity.

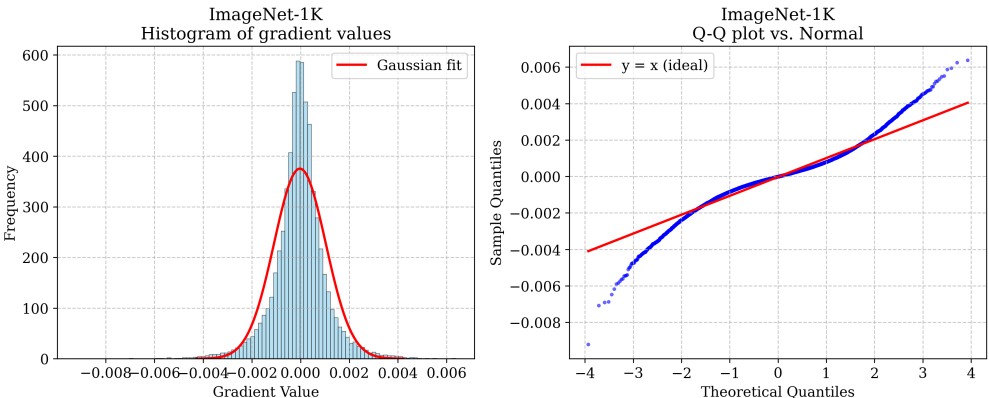

Figure 7: Illustration of InD sample gradients exhibiting sub-Gaussian behavior in each coordinate. (left) Histogram of gradient values (ImageNet-1k InD data) showing concentration around the mean with light tails, consistent with sub-Gaussianity. (right) Q-Q plot comparing empirical quantiles of InD gradients against a theoretical Gaussian distribution, confirming alignment with sub-Gaussianity.

## B    Broader Impacts and Limitations

As machine learning continues to advance, tackling the challenges of OOD detection has become essential for ensuring robust and reliable model performance in real-world applications. We introduce a median-centric framework for OOD detection that enhances OOD handling and model safety. The impact of our research goes beyond theoretical advancements, with practical applications in healthcare, autonomous systems, and finance. By improving OOD detection, we address a key challenge in model deployment, fostering greater trust and adoption of machine learning technologies. Future work could explore integrating it with generative models for outlier synthesis or adapting it to dynamic environments where OOD distributions evolve over time.

A potential limitation of Medix lies in its reliance on a mixture of unlabeled InD and OOD data, which, in practice, may be noisy, corrupted, or inconsistently sampled. However, this assumption is not unique to our method—it reflects the inherent uncertainty of real-world deployment environments and is a common starting point in robust learning theory. To make our assumptions explicit and verifiable, we rigorously formalize them in the main theorems and provide a precise geometric condition in Theorem 4.4 to justify the necessary separation required for reliable filtering. Moreover, we extend our guarantees to looser settings in Corollary 4.5, removing the need for sub-Gaussian tails and demonstrating the resilience of our framework under weaker distributional assumptions.

# C  Proof of Theorem

We summarize the notation used in this paper in Table 10. This includes definitions for gradient-related quantities, sample sizes, and filtering terms, as well as parameters that appear in our concentration bounds. Unless stated otherwise, we use boldface for vectors, $\mathcal{X}$ to denote the input space, and assume all gradients are computed with respect to a fixed pretrained model $f_\phi$.

Table 10: Notation summary. We use bold symbols for vectors and $\mathcal{X}$ to denote the input space.

| Symbol | Meaning |
|---|---|
| $\mathcal{X}$ | Input space |
| $f_\phi$ | Model parameterized by $\phi$ |
| $\ell(f_\phi(x))$ | Loss function evaluated at input $x$ |
| $\nabla \ell(f_\phi(x))$ | Gradient of loss at point $x$ |
| $\bar{\nabla}_{\text{in}}$ | Empirical mean of gradients over InD points |
| $m$ | Total number of unlabeled samples in $\mathcal{S}_{\text{wild}}$ |
| $M_{\text{in}}$ | Number of InD samples in $\mathcal{S}_{\text{wild}}$ (random under the Huber model; $\mathbb{E}[M_{\text{in}}] = (1-\pi)m$) |
| $M_{\text{out}}$ | Number of OOD samples in $\mathcal{S}_{\text{wild}}$ (random; $\mathbb{E}[M_{\text{out}}] = \pi m$) |
| $\pi \in (0,1)$ | OOD contamination ratio |
| $\mathcal{S}_{\text{wild}}$ | Unlabeled mixture of InD and OOD data |
| $\mathcal{I}$ | True InD subset of $\mathcal{S}_{\text{wild}}$ |
| $\mathcal{O}$ | True OOD subset of $\mathcal{S}_{\text{wild}}$ |
| $\mathcal{S}^\star$ | Subset selected by the ideal size-constrained EWM filter |
| $\text{ERR}_{\text{in}}$ | Fraction of InD points misclassified as OOD |
| $\text{ERR}_{\text{out}}$ | Fraction of OOD points misclassified as InD |
| $\sigma^2$ | Sub-Gaussian proxy variance of each InD gradient coordinate |
| $\sigma_{\text{out}}^2$ | Sub-Gaussian proxy variance of each OOD gradient coordinate |
| $\mu_{\text{out}}$ | Mean OOD gradient vector |
| $\Delta$ | Separation: $\|\mu_{\text{out}} - \bar{\nabla}_{\text{in}}\|_2 \geq \Delta\sqrt{d}$ |
| $\epsilon$ | Threshold for gradient deviation, $\epsilon = \sigma\sqrt{2\log(4dm/\delta)}$ |
| $\delta \in (0,1)$ | Confidence parameter for concentration bounds |
| $d$ | Dimensionality of gradient vectors |

## C.1  Bound on the Inlier Misclassification Rate

*Proof.* Write $\mathcal{I}, \mathcal{O} \subseteq \mathcal{S}_{\text{wild}}$ for the realized InD and OOD subsets, so $|\mathcal{I}| = M_{\text{in}}$, $|\mathcal{O}| = M_{\text{out}}$, and $M_{\text{in}} + M_{\text{out}} = m$.

*Reference accuracy.* The clean set $\mathcal{I}$ is feasible in equation 7 since $|\mathcal{I}| = M_{\text{in}}$, so optimality of $\mathcal{S}^\star$ gives $\|\text{EWM}(G_{\mathcal{S}^\star}) - \bar{\nabla}_{\text{in}}\|_2 \leq \|\text{EWM}(G_{\mathcal{I}}) - \bar{\nabla}_{\text{in}}\|_2$, and it suffices to bound the right-hand side. By sub-Gaussianity each centered coordinate obeys $\mathbb{P}(|G_{i,j} - \bar{\nabla}_{\text{in},j}| > \epsilon) \leq 2e^{-\epsilon^2/2\sigma^2}$, so a union bound over the at most $dm$ pairs $(i,j)$ with $i \in \mathcal{I}$, together with $\epsilon = \sigma\sqrt{2\log(4dm/\delta)}$, caps the failure probability at $\delta/2$. On the complementary event every inlier coordinate, and hence each coordinate-wise median, lies in $[\bar{\nabla}_{\text{in},j} - \epsilon, \bar{\nabla}_{\text{in},j} + \epsilon]$. Thus $\|\text{EWM}(G_{\mathcal{I}}) - \bar{\nabla}_{\text{in}}\|_\infty \leq \epsilon$, and the claim $\|\text{EWM}(G_{\mathcal{S}^\star}) - \bar{\nabla}_{\text{in}}\|_2 \leq \epsilon\sqrt{d}$ follows.

*Inlier misclassification.* Since $|\mathcal{S}^\star| = M_{\text{in}} = |\mathcal{I}|$, each inlier dropped from $\mathcal{S}^\star$ is replaced by an OOD point, so $|\mathcal{I} \setminus \mathcal{S}^\star| = |\mathcal{S}^\star \cap \mathcal{O}| \leq M_{\text{out}}$ and $\text{ERR}_{\text{in}} \leq M_{\text{out}}/M_{\text{in}}$. As $M_{\text{in}} \sim \text{Bin}(m, 1-\pi)$, set $a = 1-\pi$ and $u = \sqrt{\log(2/\delta)/2m}$; Hoeffding gives $M_{\text{in}} \geq m(a-u)$ with probability at least $1-\delta/2$, whence $M_{\text{out}}/M_{\text{in}} \leq (\pi+u)/(a-u)$. The hypothesis $m \geq 2\log(2/\delta)/(1-\pi)^2$ forces $u \leq a/2$, and using $a + \pi = 1$,

$$\frac{\pi+u}{a-u} = \frac{\pi}{a} + \frac{u}{a(a-u)} \leq \frac{\pi}{1-\pi} + \frac{2}{(1-\pi)^2}\sqrt{\frac{\log(2/\delta)}{2m}}.$$

A union bound over the two $\delta/2$ events completes the proof. □

## C.2 Bound on the Outlier Retention Rate

We now establish a non-asymptotic upper bound on the fraction of OOD points retained by the ideal MEDIX filter. This result analyzes the ideal size-constrained EWM objective; the practical greedy procedure in Algorithm 1 is used as a tractable approximation.

**Assumption C.1** (Reference EWM accuracy). Let $\mathcal{I} \subseteq \mathcal{S}_{\text{wild}}$ denote the true InD subset in the wild data. For the tolerance $\epsilon > 0$ used in Theorem 4.4, assume that

$$\left\| \text{EWM}\left(G_{\mathcal{I}}\right) - \bar{\nabla}_{\text{in}} \right\|_2 < \epsilon\sqrt{d}.$$

*Proof.* Write $\Phi(\mathcal{S}) := \|\text{EWM}(G_{\mathcal{S}}) - \bar{\nabla}_{\text{in}}\|_2$. Feasibility of $\mathcal{I}$, optimality of $\mathcal{S}^\star$, and Assumption C.1 give $\Phi(\mathcal{S}^\star) \leq \Phi(\mathcal{I}) < \epsilon\sqrt{d}$; this and $|\mathcal{S}^\star| = M_{\text{in}}$ are the only properties of $\mathcal{S}^\star$ we use.

Call $i \in \mathcal{O}$ *irregular* if $\|G_i - \mu_{\text{out}}\|_\infty > \Delta - \epsilon$; by sub-Gaussianity and a union bound over coordinates, $\mathbb{P}(i \text{ irregular}) \leq p_{\text{out}} := 2d\exp(-(\Delta - \epsilon)^2/2\sigma_{\text{out}}^2)$. With $Y$ the number of irregular OOD points and $u := \sqrt{\log(2/\delta)/2m}$, two applications of Hoeffding, on $\{G_i\}_{i \in \mathcal{O}}$ and on $M_{\text{out}} \sim \text{Bin}(m, \pi)$, each hold with probability $1 - \delta/2$:

$$\frac{Y}{M_{\text{out}}} \leq p_{\text{out}} + \sqrt{\frac{\log(2/\delta)}{2M_{\text{out}}}}, \qquad M_{\text{out}} \geq m(\pi - u).$$

The hypothesis $m \geq 2\log(2/\delta)/\pi^2$ forces $u \leq \pi/2$, so $\pi - u \geq \pi/2$ and the count bound sharpens the other two estimates to

$$\sqrt{\frac{\log(2/\delta)}{2M_{\text{out}}}} \leq \sqrt{\tfrac{2}{\pi}}\, u, \qquad \frac{M_{\text{in}}}{M_{\text{out}}} \leq \frac{1 - \pi + u}{\pi - u} \leq \frac{1 - \pi}{\pi} + \frac{2u}{\pi^2}.$$

Suppose $a := |\mathcal{S}^\star \cap \mathcal{O}|$ exceeded $Y + M_{\text{in}}/2$. Then $\mathcal{S}^\star$, of size $M_{\text{in}}$, would hold more than $M_{\text{in}}/2$ regular OOD points, so in every coordinate its median would lie within $\Delta - \epsilon$ of $\mu_{\text{out},j}$, i.e. $\|\text{EWM}(G_{\mathcal{S}^\star}) - \mu_{\text{out}}\|_2 \leq (\Delta - \epsilon)\sqrt{d}$. With the separation $\|\mu_{\text{out}} - \bar{\nabla}_{\text{in}}\|_2 \geq \Delta\sqrt{d}$ the triangle inequality would force $\Phi(\mathcal{S}^\star) \geq \epsilon\sqrt{d}$, contradicting the opening bound. Hence $a \leq Y + M_{\text{in}}/2$, and dividing by $M_{\text{out}}$,

$$\text{ERR}_{\text{out}} = \frac{a}{M_{\text{out}}} \leq p_{\text{out}} + \sqrt{\tfrac{2}{\pi}}\, u + \frac{1 - \pi}{2\pi} + \frac{u}{\pi^2}.$$

Since $\sqrt{2/\pi} \leq \pi^{-2}$ on $(0, 1/2]$, the two $u$-terms collapse to $2u/\pi^2$; substituting $u$ and $p_{\text{out}}$ and collecting the two $\delta/2$ failures by a union bound gives the claim. □

## C.3 Reference EWM Accuracy without Sub-Gaussianity

*Proof.* Fix a coordinate $j$ and write $\mu_j := \bar{\nabla}_{\text{in},j}$. Markov applied to the fourth moment gives, for each $i \in \mathcal{I}$,

$$\mathbb{P}(G_{i,j} - \mu_j > \epsilon) \leq \mathbb{P}(|G_{i,j} - \mu_j| > \epsilon) \leq \frac{\mu_4}{\epsilon^4} =: p_\epsilon,$$

and the same for the lower tail. The coordinate-wise median exceeds $\mu_j + \epsilon$ only if at least half the $G_{i,j}$ do, that is, only if the i.i.d. indicators $\mathbf{1}\{G_{i,j} - \mu_j > \epsilon\}$, each of mean at most $p_\epsilon < 1/2$, average to at least $1/2$. Hoeffding bounds that by $\exp(-2M_{\text{in}}(\frac{1}{2} - p_\epsilon)^2)$, and the symmetric argument handles the lower deviation, so

$$\mathbb{P}(|\text{EWM}_j(G_{\mathcal{I}}) - \mu_j| > \epsilon) \leq 2\exp\left(-2M_{\text{in}}(\tfrac{1}{2} - p_\epsilon)^2\right).$$

A union bound over the $d$ coordinates leaves, with probability at least $1 - 2d\exp(-2M_{\text{in}}(\frac{1}{2} - p_\epsilon)^2)$, every coordinate within $\epsilon$ of its target, so $\|\text{EWM}(G_{\mathcal{I}}) - \bar{\nabla}_{\text{in}}\|_2 \leq \epsilon\sqrt{d}$. Substituting $p_\epsilon = \mu_4/\epsilon^4$ finishes the proof. □

