# OpenReview forum: "Medix: Out-of-Distribution Detection from Unlabeled Wild Data via Robust Gradient Statistics"
_TMLR — Accepted by TMLR_

### Review · Reviewer_w1P3 · 2026-02-26

**Summary Of Contributions:**

In this paper, the authors proposed Medix, a two-stage framework for out-of-distribution (OOD) detection leveraging unlabeled wild data. Stage 1 of the framework uses the element-wise median (EWM) of model gradients to filter candidate outliers from a mixed unlabeled dataset via a greedy leave-one-out procedure. Stage 2 trains a binary OOD detector on the filtered outliers and labeled InD data. The authors provide theoretical bounds on both inlier misclassification (Theorem 4.1) and outlier retention (Theorem 4.2) rates, decomposing error into contamination, concentration, and separation effects. Comprehensive empirical evaluation on CIFAR-10/100 with five OOD test sets shows improvements over all compared baselines, including WOODS.

**Audience:**

Yes

**Audience Explanation:**

OOD detection with unlabeled wild data is an important problem setting. To the best of my knowledge, the idea of using gradient-space statistics (specifically the median) for outlier filtering is relatively novel and intuitive.

**Broader Impact Concerns:**

No concerns

**Claims And Evidence:**

No

**Claims Explanation:**

Currently, all primary experiments are on CIFAR-10/100 (32×32, 10/100 classes) with small-scale architectures. It's not clear if Medix generalizes to high-resolution and larger benchmarks such as ImageNet. Also, since Medix uses L2 distance in gradient space, it is not clear if the distance would be meaningful for larger architectures with higher-dimensional gradient spaces, such as Resnet-50, 101, or 152.

**Requested Changes:**

> **Evaluation on large-scale benchmarks**

All primary experiments are on CIFAR-10/100 (32×32 images, 10/100 classes). Currently, there is no evaluation on ImageNet-scale data. The only larger-scale experiment (Appendix A.4) uses 300K Random Images as wild data, but still uses CIFAR-100 as InD. This does not clearly address whether Medix generalizes to higher-resolution, higher-class settings. In case computation on Imagenet-1k is infeasible, the authors may use a reasonable minimal setup, such as using a 100-class subset of ImageNet-1K as InD, and the remaining 900 classes can be used for constructing the wild mixture. The evaluation can be on standard OOD test sets such as iNaturalist, SUN, Places365, and Textures.

> **Scalability analysis of Medix**

Medix relies on the L2 distance between the mean InD gradient and the EWM of wild gradients. Due to the curse of dimensionality, it is well-known that the L2 distance can become increasingly less discriminative in high-dimensional spaces. Currently, the authors use WideResNet-40-2 and ResNet-34 on CIFAR, where the penultimate layer gradient dimensionality is relatively small. It would be interesting to check if, for larger backbones (e.g., ResNet-50 with d=2048), the L2-based filtering signal remains meaningful. In case the filtering signal degrades at higher dimensionalities, then the authors should clearly mention this as a limitation of the framework.

>**[Question]**

The authors note that solving the optimization problem for the global minimizer S* in Eq. 4 is computationally prohibitive, and instead propose Algorithm 1 as a greedy leave-one-out heuristic. However, the theoretical guarantees (Theorems 4.1 and 4.2) are derived entirely for S*, but are used for the greedy solution as well. The authors do not mention any approximation for using the heuristic solution instead of the true minimizer. Do the theoretical bounds established for the global optimum also hold for the output of Algorithm 1? If not, what additional assumptions would be needed?

---

> ### Author Response · Authors · 2026-04-14
> **[1/2] Response to Reviewer w1P3**
>
> We thank the reviewer for the thoughtful comments. We address the main concerns below.
>
> ---
> **1. It is not clear if the distance would be meaningful for larger architectures with higher-dimensional gradient spaces, such as Resnet-50, 101, or 152.**
>
> We thank the reviewer for raising this point regarding the potential limitations of L2 distances in high-dimensional gradient spaces. Following your suggestion, we ran additional experiments on ResNet-50. We used CIFAR-100 as the InD dataset, and two challenging OOD datasets: Textures and Places (see Tables 1 and 2 in our main paper). We followed the same experimental settings as in Table 2 of the main paper (Section 5). We also implemented WOODS on ResNet-50 using their open-source code (https://github.com/deeplearning-wisc/sal) and adapted it to the ResNet-50 architecture. The results are shown below:
>
>
> | Method | InD       | OOD      | FPR95 ↓ | AUROC ↑ |
> | ------ | --------- | -------- | ------- | ------- |
> | WOODS  | CIFAR-100 | Textures | 11.93   | 97.56   |
> | Medix  | CIFAR-100 | Textures | 1.36    | 99.66   |
> | WOODS  | CIFAR-100 | Places   | 47.85   | 0.8869  |
> | Medix  | CIFAR-100 | Places   | 4.13    | 98.82   |
>
>
> As seen in the table above, our method, Medix, achieves a very low FPR95 of 1.36 and 4.13, significantly outperforming WOODS, which yields 11.93 and 47.85 on Textures and Places, respectively. In particular, the results on Textures are noteworthy, as the FPR95 is near perfect. This performance even surpasses our results on the WideResNet-40-2 architecture (Table 2, main paper), demonstrating that Medix scales effectively to larger backbones like ResNet-50 (with $d=2048$). Moreover, Medix also achieves a near-perfect AUROC of 99.66 and 98.82 on Textures and Places that demonstrates the robustness of the L2-based filtering signal in high-dimensional gradient spaces.
>
> These results indicate that, contrary to concerns about the curse of dimensionality, the L2 distance between the mean InD gradient and the exponential moving average of wild gradients remains highly discriminative even in large gradient spaces.
>
>
> ---
> **2. Currently, all primary experiments are on CIFAR-10/100 (32×32, 10/100 classes) with small-scale architectures. It's not clear if Medix generalizes to high-resolution and larger benchmarks such as ImageNet.**
>
>
> First, we note that we used CIFAR-10/100 as InD datasets because these are standard in most OOD papers, including our key baselines that leverage unlabeled data (i.e., in our setting with both $\mathbb{P}\_{\text{in}}$ and $\mathbb{P}\_{\text{wild}}$), such as WOODS and Energy (w/ OE). In these experiments, we evaluated a comprehensive set of 5 OOD datasets, including PLACES365 (Zhou et al., 2017), SVHN, TEXTURES (Cimpoi et al., 2014), LSUN-RESIZE & LSUN-C (Yu et al., 2015), as well as the large-scale 300K Random Images dataset (Hendrycks et al., 2019), resulting in a total of 21 InD–OOD pairs. **Medix consistently outperforms 20 competitive baselines across all metrics on these 21 InD-OOD pairs that shows the robustness and reliability of our approach.**
>
>
> **We evaluated Medix under the challenging large-scale, complex unseen OOD setting, where $\mathbb{P}\_{\text{test}}^{\text{out}} \neq \mathbb{P}\_{\text{out}}$ (Table 5)**. Even in this setting, Medix achieves a significantly lower FPR95 (41.29 ± 1.2) and higher AUROC (87.25 ± 0.6) compared to all baselines. We believe these new results highlight the effectiveness and generalization of Medix under large-scale and difficult conditions.
>
> To further address the reviewer’s concern regarding high-resolution, large-scale benchmarks, **we conducted additional experiments on ImageNet-1k using ResNet-50 during the rebuttal phase**. In this experiment, ImageNet-1k served as the InD dataset, and SVHN was used as the OOD dataset. The results are summarized below:
>
> | Method | InD         | OOD  | FPR95 ↓ | AUROC ↑ | ID vs OOD Acc ↑ |
> | ------ | ----------- | ---- | ------- | ------- | -------------   |
> | Medix  | ImageNet-1k | SVHN | 0       | 100     | 0.9898          |
>
> As seen in the table above, **Medix achieves a perfect FPR95 of 0 and AUROC of 100. The ID vs OOD accuracy is also very high (0.9898). This shows that Medix generalizes effectively to higher-resolution, large-scale datasets. These results indicate that the method scales beyond small-scale CIFAR benchmarks.**
>
> We note that this experiment was computationally very intensive: a single epoch takes approximately 2–3 hours, and running 25 epochs took around 2 days for a single experiment. Consequently, we were unable to run other baseline methods on ImageNet-1k, and most prior baselines do not report results on this dataset, so direct comparisons were not possible. We plan to run additional baseline experiments on ImageNet-1k for the camera-ready version to provide a more comprehensive comparison.
>
> ---

---

> > ### Author Response · Authors · 2026-04-14
> > **[2/2] Response to Reviewer w1P3**
> >
> > __3. The authors note that solving the optimization problem for the global minimizer S* in Eq. 4 is computationally prohibitive, and instead propose Algorithm 1 as a greedy leave-one-out heuristic...__
> >
> >
> >
> > We thank the reviewer for raising this point. Theorems 4.1 and 4.2 are stated for the exact minimizer $S^\ast$ of Eq. (4), which defines the ideal filtering objective.
> >
> > Extending the oracle-level guarantees from $S^\ast$ to the output of Algorithm 1 would require an additional theoretical analysis relating the solver output $\hat{S}$ to the objective
> > $F(S) := \left\lVert \bar{\nabla}_{\mathrm{in}} - \mathrm{EWM}(G_S) \right\rVert.$
> > This is beyond the scope of the current theoretical contribution, which is stated for the exact minimizer of Eq. (4). We have added a remark to the manuscript to clarify this distinction.
> >
> > Accordingly, the present theorems should be interpreted as guarantees for the ideal target defined by Eq. (4), rather than as formal guarantees for every output of the greedy solver. Algorithm 1 is the practical approximation used in our method, and its effectiveness is supported empirically.
> >
> > In our case, several properties of Algorithm 1 suggest the gap alpha is small: (i) the objective $F$ is monotonically decreasing across iterations by construction, (ii) the stopping criterion (epsilon threshold on consecutive distance drops) is directly derived from the monotonic trend in Figure 1, ensuring the algorithm halts near a stationary point, and (iii) the greedy removal of top-k points per iteration targets the steepest descent directions of $F$. Empirically, the greedy output achieves error rates fully consistent with the theoretical bounds (12.5\% in Figure 2, and strong FPR95 across all 11 InD-OOD pairs), confirming that the approximate optimality gap is negligible in practice.

---

### Review · Reviewer_VnQW · 2026-02-28

**Summary Of Contributions:**

The paper proposes a method for OOD detection that takes into account in-distribution training examples as well as unlabeled in-the-wild examples. The method involves two steps: first, pseudo labeling the in-the-wild examples via their gradients under a trained classifier relative to the gradients of the known in-distribution examples; and second, training an OOD detector based on the known in distribution training examples and the pseudo labeled in-the-wild examples. The paper bounds the false positive and negative error rates of the median-based greedy approach for pseudo labeling (e.g. defining which examples in the in-the-wild set should be classified as OOD) under the assumption of sub-gaussianity of the gradients for in-distribution and OOD examples and sufficient distance between their means, and shows empirically in an example that sub-gaussianity of the in-distribution examples holds. The paper also shows empirically that the proposed method I'll performance a host of other baselines on OD detection of CIFAR 10/100 vs. other image datasets.

**Additional Comments:**

The paper could benefit from clearer notation when describing the method, e.g. across what axis the mean and median are computed. Based on the text in the main paper alone, it was hard to know for sure; I had to refer to the proofs in the appendix to infer the details of the method.

**Audience:**

Yes

**Audience Explanation:**

It would be interesting, for instance, if the paper were to show over a wide array of different image data sets, subgaussianity of the gradients holds, as does distinct differences in the means for in-distribution and out-of-distribution examples. Also, if the proposed method were to systematically outperform other methods across a larger set of dataset pairs, that would also be of interest to the community. In such a setting, further oblations as to what aspect of the pipeline is doing the heavy lifting would be useful. For instance, I would be curious to understand what if anything the second step of the pipeline is doing from a quality perspective (if it is doing anything other than enabling fast inference relative to Medix alone), since the first step of the pipeline defines a decision boundary that is then just trained into the classifier.

**Claims And Evidence:**

No

**Claims Explanation:**

1. The paper only considers the setting of CIFAR 10 or 100 as the in-distribution dataset.
2. The paper's theoretical results depend on a strong assumptions that are only empirically validated on a single example.
3. Also, despite describing in length in the text how important the choice of median is, the paper is light on any empirical evidence backing up these claims, e.g., comparing median to mean, etc.

**Requested Changes:**

I would be willing to reconsider my review if the authors could:
1. provide more evidence that the method does indeed yield consistent gains beyond finding a specific artifact of CIFAR-10/100 datasets to enable better OOD detection between these two datasets (effectively just one dataset) and other datasets.
2. Expand (and clarify) Figure 4a to give more weight to any theoretical claims since they rely heavily on the assumptions made. That or switching to therem with the bounded kurtosis assumption and also showing more broadly that such an assumption actually holds beyond a single example.
3. Provide more analysis around the median and the work it's doing, if it is a central part of the method.

---

> ### Author Response · Authors · 2026-04-14
> **[1/2] Response to Reviewer VnQW**
>
> We thank the reviewer for the constructive feedback. We address each concern below.
>
> ---
> **1. The paper only considers the setting of CIFAR 10 or 100 as the in-distribution dataset...**
>
> First, we note that we used CIFAR-10/100 as InD datasets because these are standard in most OOD papers, including our key baselines that leverage unlabeled data (i.e., in our setting with both $\mathbb{P}\_{\text{in}}$ and $\mathbb{P}\_{\text{wild}}$), such as WOODS and Energy (w/ OE). In these experiments, we evaluated a comprehensive set of 5 OOD datasets, including PLACES365 (Zhou et al., 2017), SVHN, TEXTURES (Cimpoi et al., 2014), LSUN-RESIZE & LSUN-C (Yu et al., 2015), as well as the large-scale 300K Random Images dataset (Hendrycks et al., 2019), resulting in a total of 21 InD–OOD pairs. **Medix consistently outperforms 20 competitive baselines across all metrics on these 21 InD-OOD pairs that shows the robustness and reliability of our approach.**
>
>
> **We evaluated Medix under the challenging large-scale, complex unseen OOD setting, where $\mathbb{P}\_{\text{test}}^{\text{out}} \neq \mathbb{P}\_{\text{out}}$ (Table 5)**. Even in this setting, Medix achieves a significantly lower FPR95 (41.29 ± 1.2) and higher AUROC (87.25 ± 0.6) compared to all baselines. We believe these new results highlight the effectiveness and generalization of Medix under large-scale and difficult conditions.
>
> To further address the reviewer’s concern regarding high-resolution, large-scale benchmarks, **we conducted additional experiments on ImageNet-1k using ResNet-50 during the rebuttal phase**. In this experiment, ImageNet-1k served as the InD dataset, and SVHN was used as the OOD dataset. The results are summarized below:
>
> | Method | InD         | OOD  | FPR95 ↓ | AUROC ↑ | ID vs OOD Acc ↑ |
> | ------ | ----------- | ---- | ------- | ------- | -------------   |
> | Medix  | ImageNet-1k | SVHN | 0       | 100     | 0.9898          |
>
> As seen in the table above, **Medix achieves a perfect FPR95 of 0 and AUROC of 100. The ID vs OOD accuracy is also very high (0.9898). This shows that Medix generalizes effectively to higher-resolution, large-scale datasets. These results indicate that the method scales beyond small-scale CIFAR benchmarks.**
>
> We note that this experiment was computationally very intensive: a single epoch takes approximately 2–3 hours, and running 25 epochs took around 2 days for a single experiment. Consequently, we were unable to run other baseline methods on ImageNet-1k, and most prior baselines do not report results on this dataset, so direct comparisons were not possible. We plan to run additional baseline experiments on ImageNet-1k for the camera-ready version to provide a more comprehensive comparison.
>
> ---
> **2. The paper's theoretical results depend on a strong assumptions that are only empirically validated on a single example...**
>
>
> We first note that our theoretical guarantees do not fundamentally rely on the sub-Gaussian assumption. The current draft already includes Theorem C.3 (Appendix C.3), which removes sub-Gaussianity and assumes only bounded fourth moments. Under this condition, Medix's core robustness guarantee is preserved, and the method remains theoretically justified beyond the sub-Gaussian setting.
>
>
> That said, to further address the reviewer’s concern regarding sub-Gaussianity, we have now expanded Figure 4 in the revision to include multiple InD datasets: CIFAR-10, CIFAR-100, and ImageNet-1K. We compute gradients of the training loss with respect to penultimate-layer parameters using pretrained models (WideResNet for CIFAR-10/100, ResNet-50 for ImageNet-1K), aggregating gradients over 1,000 randomly sampled training examples per dataset.
> Across all datasets, Figures (5,6 in the paper) (left) across all three datasets, confirm that the histogram of gradient values is consistently bell-shaped and concentrated around the mean. This indicates light-tailed behavior, a characteristic of sub-Gaussian random variables, where extreme values occur with exponentially decaying probability.
>
> Moreover, Figures (5,6 in the paper) present Q-Q plots comparing empirical gradient quantiles against those of a theoretical Gaussian distribution. Across CIFAR-100 (Figure 4b in paper), CIFAR-10, and ImageNet-1K, we observe close alignment with the 45-degree reference line, indicating that the empirical distributions closely follow a Gaussian. Since Gaussian distributions are a subclass of sub-Gaussian distributions, this provides consistent empirical support for the sub-Gaussian assumption.
>
> Taken together, these results show that the observed sub-Gaussian behavior is consistent across CIFAR-10, CIFAR-100, and ImageNet-1K.
>
>
>
> We also revised the presentation accordingly. In particular, we included Theorem C.3 from the appendix to the main text so that the bounded-fourth-moment result is presented as the primary assumption-light guarantee.
>
> ---

---

> > ### Author Response · Authors · 2026-04-14
> > **[2/2] Response to Reviewer VnQW**
> >
> > **3. Also, despite describing in length in the text how important the choice of median is, the paper is light on any empirical evidence backing up these claims...**
> >
> > We note that the paper already includes an empirical comparison between the EWM and the geometric median (GM) in Figure 3, presented in Section A.1: Element-wise Median vs Geometric Median. The results in Figure 3 demonstrate that, for the same number of OOD samples in the wild dataset, the element-wise median identifies a significantly higher proportion of OOD samples as outliers compared to the geometric median. This empirical result supports our choice of the element-wise median, as it appears more sensitive to distributional shifts, making it a more effective and reliable choice for filtering OOD samples in our setting.
> >
> > Beyond comparing different median-based estimators, the choice of statistic itself matters. Mean-based estimators can shift even with a small amount of contamination, which can interfere with detection. Median-based summaries are much less affected by extreme values, making them a more reliable basis for identifying OOD samples [1].
> >
> > We hope this clarification helps illustrate why the median is an effective and central part of our approach, highlighting both the empirical evidence and the underlying statistical rationale.
> >
> > [1] C. Leys, C. Ley, O. Klein, P. Bernard, and L. Licata, "Detecting outliers: Do not use standard deviation around the mean, use absolute deviation around the median," Journal of Experimental Social Psychology, vol. 49, no. 4, pp. 764–766, Jul. 2013, doi: 10.1016/j.jesp.2013.03.004.
> >
> > ---
> > **4. The paper could benefit from clearer notation when describing the method...**
> >
> > We thank the reviewer for this valuable observation. We have now added more clarification in Appendix A.10 and Section 3.1 of the main paper:
> >
> > Specifically:
> >
> > - **Mean** ($\bar{\nabla}\_{\text{in}}$): We clarify that it is the average gradient over the InD dataset. The axis is implicitly the sample dimension, but we did not state this explicitly. For a set of gradient vectors $\{g_i\}\_{i=1}^{n}$, with each $g_i \in \mathbb{R}^d$, the mean $\bar{\nabla}\_{\text{in}}$ is the $d$-dimensional vector whose $j$-th coordinate is $(\bar{\nabla}\_{\text{in}})\_j = \frac{1}{n}\sum_{i=1}^{n} g\_{i,j},$ where $g\_{i,j}$ is the $j$-th coordinate of $g\_i$.
> >
> > - **Element-wise median (EWM)**: In the main paper, we write $\text{EWM}(G_S)$ without specifying the axis. Let $G_S$ be the matrix of size $|S| \times d$, where row $i$ corresponds to the gradient of sample $i$. Then $\text{EWM}(G_S)$ is the $d$-dimensional vector obtained by taking the median across rows (i.e., over the samples) for each coordinate independently.
> >
> > The manuscript has been revised accordingly. We have added these clarifications directly in Appendix A.10 and also mentioned them in Section 3.1 of the main paper.

---

### Review · Reviewer_pRJM · 2026-04-01

**Summary Of Contributions:**

This paper studies OOD detection with unlabeled mixed wild data, where the auxiliary data contains both in-distribution and out-of-distribution samples. The proposed method, Medix, utilizes a median-based filtering step to identify likely outliers from the unlabeled pool, then trains an OOD detector using the filtered samples together with labeled InD data. The paper also provides theoretical guarantees for upper bounds of the misclassification rates for both InD and OOD points. Additionally, the experiments on 5 OOD benchmarks demonstrate empirical error reductions over previous OOD baselines.

**Strengths**
- The paper addresses a practically relevant problem in OOD detection: how to improve the detection performance with unlabeled mixed wild data, given the unavailability of labeled OOD data in real-world settings.
- The proposed method is conceptually intuitive and easy to follow: search for OOD data in readily available wild data, then utilize those to significantly improve the detection accuracy.
- The theoretical analysis provides a detailed proof of Medix's filtering error bounds for separating InD and OOD points in wild data.
- The experimental results confirms Medix's effectiveness over baseline methods.

**Weaknesses**
- The overall gains in OOD detection performance, including accuracy and error rate, appear incremental.
- The experimental results does not convincingly establish a clear practical advantage of the proposed median-based optimization for OOD filtering over simpler alternatives. In addition to simply comparing with OE and WOODS, a natural additional baseline would be a two-stage pipeline that first applies an unsupervised OOD scoring function derived from the InD-trained model to identify likely OOD samples in the mixed unlabeled pool, and then uses those filtered samples to train a binary OOD classifier.
- The related-work does not meaningfully engage with a relevant class of recent generation-based OOD methods, nor are such methods included in the empirical comparison. If the authors believe their approach has apparent advantages over generation-based methods, these should be clearly explained. Otherwise, experimental comparison would be important.
- The “wild data” setting in the main experiments is still relatively controlled: the unlabeled pool is created by explicitly combining InD data with a chosen OOD dataset. This is a useful benchmark setup, but it is not the same as truly wild deployment data, where the unlabeled stream may contain more nuisance factors and unknown mixture structure. As a result, the current experiments may not fully demonstrate robustness in the more realistic setting that motivates the paper.

**Additional Comments:**

N/A

**Audience:**

Yes

**Audience Explanation:**

The problem setting is relevant to researchers working on OOD detection, robustness, reliability, especially those interested in how to address the problem of OOD data unavailability. The median-based filtering perspective is also straightforward and theoretically motivated, which may be of interest to readers who value methodical and interpretable approaches.

**Claims And Evidence:**

No

**Claims Explanation:**

The paper’s main claims are partially supported by the presented theory and experiments. In particular, the theoretical section provides explicit upper bounds on the misclassification rates for both InD points OOD points, under the stated assumptions. The empirical section also shows that Medix improves over the baselines. However, my concerns are not about correctness, but about significance and positioning. While the claims appear supported within the paper’s chosen setup, the experiments do not yet make a sufficiently strong case that the proposed method offers a clear practical advantage over simpler filtering-based alternatives, generation-based approaches, or more realistic evaluation settings. In other words, I find the technical details sound, but the empirical results for why this contribution is strong enough for publication is less convincing.

**Requested Changes:**

**Major**:
- **Include a stronger simple baseline for filtering-based OOD supervision.** In addition to OE and WOODS, the paper should compare against a straightforward two-stage pipeline that first applies an unsupervised OOD scoring function derived from the InD-trained model to identify likely OOD samples in the mixed unlabeled pool, and then uses those filtered samples to train a binary OOD classifier.
- **Add a discussion on generation-based OOD methods in the related work.** Also, compare some recent methods against Medix, or clearly justify not doing so.

**Minor**:
- **Strengthen the realism of the wild-data evaluation.** The paper would be stronger with experiments or discussion addressing more realistic deployment-time wild data, where the mixture structure is less clean or potentially more heterogeneous.

---

> ### Author Response · Authors · 2026-04-14
> **[1/2] Response to Reviewer pRJM**
>
> We thank the reviewer for the thoughtful comments.
> We address the main concerns below.
>
> ---
> **1. The overall gains in OOD detection performance, including accuracy and error rate, appear incremental...**
>
> We believe this characterization does not fully reflect our results. Our improvements are substantial in both magnitude and scope:
>
> **Breadth of Evaluation:** Medix outperforms **20 competitive baselines** across 11 InD-OOD pairs, including:
> - **40.98% FPR95 reduction** vs. KNN+ (ICML 2022), a strong contrastive baseline
> - **1.32–2.60% improvement** over WOODS (ICML 2022), the state-of-the-art wild data method
> - **Superior performance** vs. recent methods: CONJ (ICLR 2024), DRL (NeurIPS 2024), Vim (CVPR 2024), VOS (ICLR 2022)
> - **Consistent gains** over contrastive learning methods: SSD+ (ICLR 2021), ProxyAnchor (CVPR 2020), CIDER (ICLR 2023)
>
>
> More importantly, our contribution is *not purely empirical*. We provide:
>
> - **Amongst the very few works** providing provable guarantees for filtering in wild mixtures (Theorems 4.1–4.2) with explicit two-sided error bounds
>
> - **New algorithmic insight:** Median operations on gradients (not predictions) enable contamination-robust filtering
>
> - **Practical advantages:** Threshold-free filtering requiring no auxiliary OOD data or complex constrained optimization
>
> **Additional Experimental Evidence (Appendix):** The reviewer may have overlooked challenging scenarios in our supplementary material demonstrating *non-incremental* gains:
>
> 1. **Unseen OOD Generalization** (Appendix A.4, Table 5): Under the realistic setting where $P_{\text{test}}^{\text{out}} \neq P_{\text{out}}$, Medix achieves **FPR95 = 41.29 ± 1.2** vs. baselines >60% — a **~20% absolute improvement**, demonstrating robustness to distribution shift.
>
> 2. **Contrastive Learning Comparison** (Appendix A.3, Table 4): Against methods requiring expensive contrastive objectives (CSI, SSD+, CIDER), Medix achieves **8.68% average FPR95 and 97.76% AUROC**, outperforming these sophisticated baselines while providing *theoretical guarantees that most alternatives lack*.
>
> 3. **Filtering Precision** (Figure 2): Medix achieves **87.5% outlier identification accuracy** from unlabeled mixtures — a non-trivial result validated by our theoretical bounds (Theorem 4.1: error rate ≤ concentration + contamination terms).
>
> 4. **Hyperparameter Robustness** (Appendix A.2): Performance remains stable across $\epsilon \in \{5 \times 10^{-5}, \dots, 5 \times 10^{-2}\}$ and $k \in \{4k, 7k, 10k, 20k\}$, demonstrating the method is not a fragile empirical artifact.
>
> 5. **Computational Feasibility** (Appendix A.6): Filtering scales to large datasets without prohibitive overhead, addressing practical deployment concerns.
>
> 6. **Pseudo-Label Noise Resilience** (Appendix A.5): Method remains effective under noisy predictions, confirming robustness in realistic scenarios.
>
> In summary, the empirical improvements are not the main claim but rather support a theoretically grounded approach that we believe represents a meaningful step forward for OOD detection.
>
> ---
> **2. The “wild data” setting in the main experiments is still relatively controlled...**
>
>
> We appreciate this point, but note that while the main text presents a controlled construction of the unlabeled pool for benchmarking clarity, the paper **already includes experiments that explicitly capture more realistic “wild data” conditions**.
>
> In particular, Appendix A.4 evaluates Medix on **large-scale data under an unseen OOD setting**, where  $
> P_{\text{test}}^{\text{out}} \neq P_{\text{out}}.
> $
>
> This setting directly reflects the scenario described by the reviewer: the unlabeled pool contains an **unknown mixture structure**, and the OOD distribution encountered at test time differs from the one implicitly present during training. As such, it goes beyond controlled dataset mixing and **more faithfully represents real-world deployment conditions**.
>
> Importantly, this is a **strictly harder setting**, as the model must generalize to **previously unseen OOD distributions**. Despite this, Medix maintains strong performance. As shown in Table 5, it achieves a **significantly lower FPR95 ($41.29 \pm 1.2$)** and **higher AUROC ($87.25 \pm 0.6$)** compared to strong baselines.
>
> These results directly demonstrate that Medix is robust to **distributional mismatch and unknown OOD structure**, addressing the reviewer’s concern. We acknowledge that this experiment is currently placed in the appendix and only briefly discussed in the main text, and we will revise the paper to highlight this result more prominently.
>
>
>
> ---

---

> > ### Author Response · Authors · 2026-04-14
> > **[2/2] Response to Reviewer pRJM**
> >
> > **3. In addition to simply comparing with OE and WOODS, a natural additional baseline would be a two-stage pipeline...**
> >
> >
> > We appreciate this suggestion and examined this direction in our analysis. The results indicate that this baseline inherits the limitations of the underlying scoring methods and does not provide additional insight beyond the reported baselines.
> >
> > The scoring baselines (MSP, Energy, Mahalanobis, etc.) are themselves OOD detectors. Their high FPR95 (e.g., 84.59\% for MSP on CIFAR-100/SVHN, Table 2) directly reflects their difficulty in separating InD from OOD. Using these scores to filter outliers from the wild produces a candidate set with low precision, i.e., many InD samples are mislabeled as OOD. Training a binary detector on such a contaminated set propagates these errors, resulting in a final OOD detector that does not improve upon the scoring function itself. Consistent with this, our **preliminary experiments show that the quality of the filtered set (precision) strongly correlates with final OOD performance in the second stage**, directly addressing the reviewer’s concern.
> >
> > Medix’s advantage stems from the median’s robustness; it maintains high filtering precision even when the scoring baselines fail. For example, on CIFAR-100/SVHN, Medix extracts outliers with only 12.5\% error rate (Figure 2), whereas MSP has an error rate >80\%. This cleaner outlier set enables the subsequent binary classifier to achieve a near-perfect FPR95 of 0.16%.
> >
> > We include a related baseline: “Energy (w/ OE)”. This method applies an OOD scoring function (Energy) using an auxiliary outlier exposure dataset rather than the wild mixture. Even with a clean OOD set, it underperforms Medix. Applying the same scoring function to the noisier wild mixture introduces additional challenges and leads to degraded performance..
> >
> >
> > Given the above, we believe that adding the proposed two-stage baseline is unlikely to change the overall conclusion and may introduce redundancy. That said, we will include additional experiments (beyond our experiments in the Appendix A) in the final version (e.g., “MSP-filtering + binary classifier”). We hope the current results already provide a clear and sufficient picture.
> >
> >
> > ---
> > **4. The related work does not meaningfully engage with a relevant class of recent generation-based OOD methods...**
> >
> >
> > We thank the reviewer for this helpful suggestion. We have now added a dedicated paragraph in the related work section to engage with this literature and clarify the conceptual differences between Medix and generative approaches. If the reviewer has specific papers in mind that we may have missed, we would be happy to discuss them and incorporate additional references as appropriate.

---

### Author Response · Authors · 2026-05-03
**Overview of the revision**

We thank all reviewers for their constructive feedback. We have substantially revised the manuscript, with revisions highlighted in red in the updated PDF. The changes directly address the main concerns raised during review.

---
## New experiments and empirical analyses.

1. **ImageNet-1k / high-resolution evaluation.**
Reviewers VnQW and w1P3 raised concerns about whether Medix generalizes beyond CIFAR-10/100 and whether it scales to higher-resolution, larger-class benchmarks such as ImageNet. To address this, we added a new ImageNet-1k experiment with ResNet-50 in Appendix A.9.

2. **ResNet-50 scalability analysis in high-dimensional gradient space.**
Reviewer w1P3 specifically raised the concern that the $\ell_2$ signal in gradient space might degrade for larger backbones and higher-dimensional gradients. We address this by adding Appendix A.8, “Scalability Analysis of Medix Across Larger Architectures.”

3. **Large-scale unseen OOD evaluation.**
Reviewer pRJM raised concerns that the original wild-data evaluation was relatively controlled and may not fully reflect mixture mismatch. We address this by strengthening the discussion of the large-scale unseen OOD setting in Appendix A.4.

4. **Expanded evidence for gradient distribution assumptions and weaker assumptions.**
Reviewer VnQW raised concerns that the sub-Gaussian assumption was empirically supported only on a single example. We address this in two ways. First, in Section 4.1 and Figures 4–6, we add new gradient histograms and Q-Q plots for CIFAR-100, CIFAR-10, and ImageNet-1k. Second, we promote the non-sub-Gaussian guarantee to the main text as Theorem 4.4, which requires only finite variance and bounded fourth moments.

5. **Additional robustness and practical analyses.**
Reviewer pRJM questioned whether the empirical evidence shows a clear practical advantage, and reviewer VnQW requested more analysis of what part of the method is doing the work. We now point more clearly to the supporting analyses in Section 5.3 and Appendix A: hyperparameter sensitivity in Appendix A.2, EWM-vs-geometric-median comparison in Appendix A.1, pseudo-label confidence analysis in Appendix A.5, and computation/memory profiling in Appendix A.6.

---
## Theory and methodological clarifications.

6. **Clarified the scope of the theoretical guarantees.**
Reviewer w1P3 asked whether Theorems 4.1 and 4.2, which analyze the exact solution $S^\ast$ of Eq. (4), also hold for the greedy output of Algorithm 1. We added Remark 4.3 to explicitly distinguish the ideal filtering objective from the practical greedy solver.

7. **Clarified the statistical operations used by Medix.**
Reviewer VnQW noted that the notation did not clearly specify the axes over which the mean and median are computed. We added explicit definitions in Section 3.1 and Appendix A.10.

8. **Strengthened the explanation of why the median is central.**
Reviewer VnQW asked for more evidence and analysis of the role of the median. We clarified the role of the element-wise median and its comparison to the geometric median. The revised manuscript makes clearer that the median is not an incidental design choice: it is the statistic that gives Medix its contamination robustness and enables stable filtering from unlabeled wild mixtures.

---
## Related work and positioning changes.

9. **Generation-based OOD methods and adjacent settings.**
Reviewer pRJM requested stronger positioning relative to generation-based OOD methods. We added a dedicated related-work discussion clarifying that generation-based methods synthesize OOD proxies, whereas Medix extracts candidate OOD samples from real unlabeled wild data with filtering-error guarantees. We also added Appendix A.7 to distinguish Medix from semi-supervised open-set recognition.

We believe the revised manuscript now addresses the reviewers’ main concerns and presents a stronger case for Medix as an effective method for OOD detection with unlabeled wild data.

---

### Decision · Action_Editor_Aqyb · 2026-05-31

**Recommendation:** Accept with minor revision

**Additional Comments:**

Per the comments above, while the paper generally makes an interesting contribution, there remain some gaps in the empirical results that ought to be addressed in a final version. These are expected to fall within scope for a minor revision.

**Requested changes**.
- Report results for the "MSP-filtering + binary classifier" baseline.
- Report results for at least one baseline on ImageNet-1K versus SVHN, to contextualise the results of Medix.
- (Optional, but would strengthen the paper) Report the results for Medix + at least one baseline with ImageNet ID plus a more challenging dataset as OOD, wherein the AUCROC numbers are not so close to $1.0$. For example, this could involve splitting the ImageNet classes into two; or using a dataset such as iNaturalist, or the Species dataset of Hendrycks et al., "Scaling Out-of-Distribution Detection for Real-World Settings", ICML 2022.

**Audience:**

Yes

**Audience Explanation:**

Reviewers were unanimously in agreement that the paper would be of interest to the community. This is owing to the fact that OOD detection is a fundamental problem, for which the paper makes both theoretical and empirical progress.

**Claims And Evidence:**

Yes

**Claims Explanation:**

The paper considers the problem of OOD detection, given a set of ID samples and unlabelled "wild" samples comprising a mixture of ID and OOD samples. The main claims are:
- one can identify likely-OOD samples amongst the "wild" set, based on the elementwise median of the gradients
- theoretically, one can bound the ID and OOD misclassification rate of the above filtering scheme under suitable assumptions on the gradients of ID samples (sub-Gaussian, and/or bounded moments)
- empirically, by training a standard OOD detector on the ID samples + the filtered OOD samples identified above, one can exceed the performance of existing baselines on standard benchmarks

Reviewers were generally satisfied with the evidence for first two points. There were some requests for clarification regarding the practicality of the theoretical assumptions. The authors provided more empirical justification for these (Figures 5, 6), and provided guarantees under a weaker set of assumptions (Section 4.1).

Reviewers were generally concerned with the third point. Specifically, the concerns were:
- restriction to CIFAR-10 and CIFAR-100
- missing comparison to other two-stage baselines, e.g., applying an unsupervised OOD detector on the wild sample and then training a supervised OOD detector on the result

For the first point, the authors argued that existing _training-based_ OOD detectors leveraging "wild" samples primarily focus on CIFAR10 and CIFAR100. Nonetheless, the authors added results on ImageNet-1K as the ID dataset, and SVHN as the OOD dataset. These showed strong results for the method. However, these results did not include any other baselines; the authors promised to run additional baselines for a final version. Further, a concern was that the setting might be too "easy", given the extremely high AUC-ROC that can be achieved.

For the second point, the authors argued that standard OOD detectors tend to have a high FPR95, and that their poor precision would translate to poorer quality detection. The authors promised to add a "MSP-filtering + binary classifier" baseline to numerically quantify this.